# A *Cognac* SHOT TO FORGET BAD MEMORIES: CORRECTIVE UNLEARNING IN GNNS

## ABSTRACT

Graph Neural Networks (GNNs) are increasingly being used for a variety of ML applications on graph data. As graph data does not follow the independently and identically distributed (i.i.d) assumption, adversarial manipulations or incorrect data can propagate to other datapoints through message passing, deteriorating the model's performance. To allow model developers to remove the adverse effects of manipulated entities from a trained GNN, we study the recently formulated problem of *Corrective Unlearning*. We find that current graph unlearning methods fail to unlearn the effect of manipulations even when the whole manipulated set is known. We introduce a new graph unlearning method, **Cognac**, which can unlearn the effect of the manipulation set even when only 5% of it is identified. It recovers most of the performance of a strong oracle with fully corrected training data, even beating retraining from scratch without the deletion set while being 8x more efficient. We hope our work guides GNN developers in fixing harmful effects due to issues in real-world data post-training.

## 1 INTRODUCTION

Graph Neural Networks (GNNs) are seeing widespread adoption across diverse domains, from recommender systems to drug discovery (Wu et al., 2022; Zhang et al., 2022), and are being scaled to large training sets for graph foundation models (Mao et al., 2024). However, in these large-scale settings, it is prohibitively expensive to verify the integrity of every sample in the training data, which can potentially affect desiderata like fairness (Konstantinov & Lampert, 2022), robustness (Paleka & Sanyal, 2023; Günnemann, 2022) and accuracy (Sanyal et al., 2021).

Making the training process itself robust to minority populations (Günnemann, 2022; Jin et al., 2020) is challenging and can adversely affect fairness and accuracy (Sanyal et al., 2022). Consequently, model developers may want post-hoc ways to remove the adverse impact of manipulated training data if they observe problematic model behavior on specific distributions of test-time inputs. Such an approach follows the recent trend of using post-training interventions to ensure models behave in intended ways (Ouyang et al., 2022). Recently, Goel et al. (2024) formulated *Corrective Unlearning* as the challenge of removing adverse effects of manipulated data with access to only a representative subset for unlearning while being agnostic to the type of manipulations. We study this problem in the context of GNNs, which face unique challenges due to the graph structure. The traditional assumption of independent and identically distributed (i.i.d.) samples does not hold for GNNs, as they use a message-passing mechanism that aggregates information from neighbors. This process makes GNNs vulnerable to adversarial perturbations, where modifying even a few nodes can propagate changes across large portions of the graph and result in widespread changes in model predictions (Bojchevski & Günnemann, 2019b; Zügner et al., 2018). As a result, any unlearning approach for GNNs must remove the influence of manipulated entities on neighbors to be effective.

Corrective Unlearning is the problem of removing the influence of arbitrary training data manipulations on a trained model using only a representative subset of the manipulated data. In this work, we focus on the use of GNNs in node classification tasks, studying unlearning for targeted binary class confusion attacks (Lingam et al., 2024) on both edges and nodes. For edge unlearning, we evaluate the unlearning of spurious edges that change the graph topology in a way that violates the homophily assumption. For node unlearning, we utilize a label flip attack (Lingam et al., 2024) which is used as a classical graph adversarial attack, similar to the Interclass Confusion attack (Goel et al., 2022).

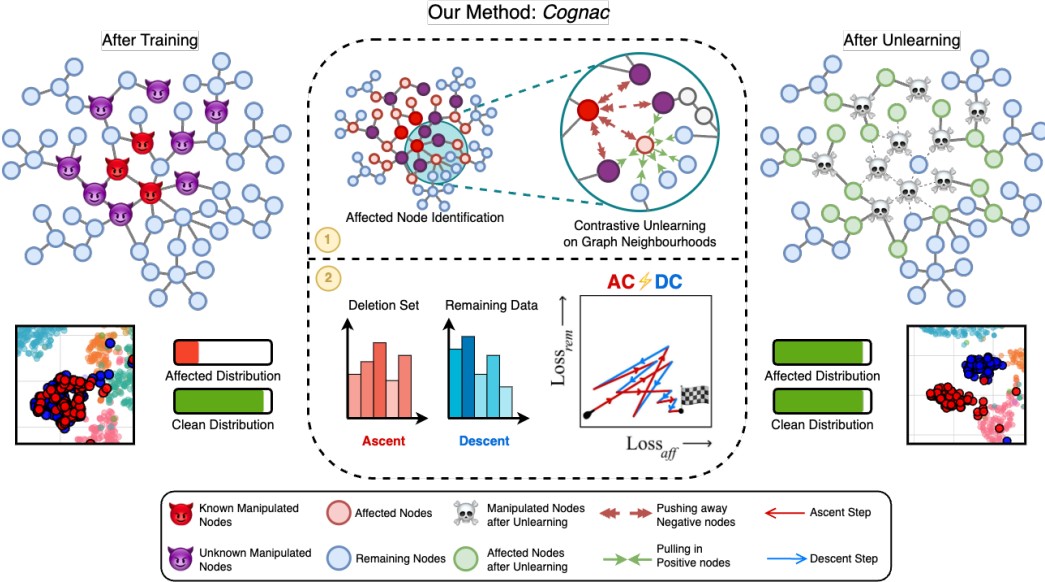

Figure 1: **Illustration of our method *Cognac*.** Initially (Left), the model is trained on manipulated data (Devils), out of which only a subset is identified for deletion (Dark-red-devils). Our method alternates between two steps. The first step (1) identifies neighbors by the deletion set, which can include both nodes from the remaining data (Light-red) and unidentified manipulated nodes (Purple), and pushes their representation away from the deletion set and towards other nodes in the neighborhood. The second step (2) performs ascent on the deletion set labels, and descent on the remaining data with separate optimizer instances. This cleanly separates the embeddings of the affected classes (Right), resulting in improved accuracy on the affected distribution, while maintaining it on the remaining distribution.

First, we evaluate whether existing GNN unlearning methods are effective in removing the impact of manipulated entities. Our findings reveal that these methods consistently fail, even when provided with the complete set of manipulated entities. We then propose our method, ***Cognac***, which unlearns by alternating between two components, as illustrated in Figure 1. The first component *Contrastive unlearning on Graph Neighborhoods* (***CoGN***), finds affected neighbors of the known deletion set, updating the GNN weights using a contrastive loss that pushes representations of the affected neighbors away from the deletion entities while staying close to other neighbors. The second component, *AsCent DesCent de♮coupled* (***AC♮DC***) applies the classic i.i.d unlearning method of gradient ascent on the deletion set and gradient descent on the retain set. We use separate optimizers for these components as we find it essential for stable dynamics when optimizing the competing objectives.

Our proposed method shows promising results, recovering most of the performance of an oracle model with access to a clean label version of the full graph, even if only 5% of the manipulated entities are identified. This is the first corrective unlearning method that succeeds in removing adversarial class confusion with a small fraction of the manipulated set. Finally, we perform a detailed analysis with ablations, where we identify an interesting tradeoff specific to graph corrective unlearning. Particularly, using a larger fraction of the manipulation set for deletion while helpful for unlearning, can remove more structural information from the graph. Keeping the manipulated entities but removing the effect of their features and labels helps our method match Oracle performance. Overall, *Cognac* offers users of GNNs an efficient, highly effective post-training strategy to remove adverse effects of manipulated data.

## 2 RELATED WORK

Graph-based attacks, such as Sybil (Douceur, 2002) and link spam farms (Wu & Davison, 2005), have long affected the integrity of social networks and search engines by exploiting the trust in-

herent in node identities and edge formations. Recent works reveal that even state-of-the-art GNN architectures are vulnerable to simple attacks on the trained models which either manipulate existing edges and nodes or inject new adversarial nodes (Sun et al., 2019; Dai et al., 2018; Zügner & Günnemann, 2019; Geisler et al., 2024). Parallelly, works have characterized the influence of specific nodes and edges that can guide such attacks (Chen et al., 2023). One strategy to mitigate the influence of such attacks is robust pretraining, such as using adversarial training (Yuan et al., 2024; Zhang et al., 2023). Post-hoc interventions like unlearning act as a complementary layer of defense, helping model developers when attacks slip through and still adversely affect a trained model.

Removing the impact of manipulated entities begins with their identification (Brodley & Friedl, 1999), for which multiple strategies exist like data attribution (Ilyas et al., 2022), adversarial detection, and automated or human-in-the-loop anomaly detection (Northcutt et al., 2021). Once identified, various approaches have been proposed to mitigate specific effects of the manipulated data, including model debiasing (Fan et al., 2024) and concept erasure (Belrose et al., 2023). While these approaches have similar goals, that is, post-hoc removing undesirable effects of corrupted training data, unlearning attempts to do this without precise knowledge of the nature of corruptions and its effects. This is useful in adversarial settings where effects can be obfuscated, and harm multiple desiderata simultaneously (Paleka & Sanyal, 2023).

Machine unlearning gained initial interest for privacy applications to serve user data deletion requests (Nguyen et al., 2022). *Exact Unlearning* procedures remove or retrain parts of the ML system that *saw* the data to be deleted, guaranteeing perfect unlearning by design (Chen et al., 2022b;a; Bourtoule et al., 2021). However, they can incur exponential costs with sequential deletion requests (Warnecke et al., 2023). Therefore, we focus on *Inexact Unlearning* methods, which either provide approximate guarantees for simple models (Chien et al., 2022; Wu et al., 2023b) or like us, empirically show unlearning through evaluations for deep neural networks (Wu et al., 2023a; Cheng et al., 2023; Li et al., 2024b). Due to the non-i.i.d nature of graphs, GNN unlearning methods need to remove the effects of deletion set entities on the remaining entities, which distinguishes the subdomain of Graph Unlearning (Said et al., 2023).

Recently, machine unlearning has received newfound attention beyond privacy applications (Pawelczyk et al., 2024; Schoepf et al., 2024; Li et al., 2024a). Goel et al. (2024) demonstrated the distinction between the *Corrective* and *Privacy-oriented* unlearning settings for i.i.d classification tasks, emphasising challenges when not all manipulated data is identified for unlearning. In this work, we focus on the intersection of corrective unlearning for graphs, evaluating existing methods and making significant progress through our proposed method *Cognac*.

## 3  CORRECTIVE UNLEARNING FOR GRAPH NEURAL NETWORKS

We now formulate the corrective unlearning problem for graph-structured, non-i.i.d data. We consider a graph $\mathcal{G} = (\mathcal{V}, \mathcal{E})$, where $\mathcal{V}$ and $\mathcal{E}$ represents the constituent set of nodes and edges respectively. For each node $\mathcal{V}_i \in \mathcal{V}$, there is a corresponding feature vector $\mathcal{X}_i$ and label $\mathcal{Y}_i$, with $\mathcal{V} = (\mathcal{X}, \mathcal{Y})$. Consistent with prior work in unlearning on graphs (Wu et al., 2023a; Li et al., 2024b), we focus on semi-supervised node classification using GNNs. GNNs use the message-passing mechanism, where each node aggregates features from its immediate neighbors. The effect of this aggregation process propagates through multiple successive layers, effectively expanding the receptive field of each node with network depth. This architecture inherently exploits the principle of homophily, a common property in many real-world graphs where nodes with similar features or labels are more likely to be connected than not.

While assuming homophily is extremely useful for learning representations from graph data, annotation mistakes or adversarial manipulations that create dissimilar neighborhoods or connect otherwise dissimilar nodes can easily harm the learned representations (Zügner & Günnemann, 2019). This motivates our study of post-hoc correction strategies like unlearning for GNNs. Following Goel et al. (2024), we adopt an adversarial formulation that subsumes correcting more benign mistakes.

**Adversary's Perspective.** The adversary can reduce model accuracy on a target distribution by manipulating parts of the clean training data $\mathcal{G}$. This can be done in the following ways: (1) adding spurious edges $\hat{\mathcal{E}}$, resulting in $\mathcal{E}' = \mathcal{E} \cup \hat{\mathcal{E}}$; or (2) manipulating node information, $\mathcal{V}' = f_m(\mathcal{V})$, where $f_m$ manipulates a subset of nodes by changing their features or labels. We define $S_m$ as the

set of manipulated entities, either the manipulated subset of nodes or the added spurious edges $\hat{\mathcal{E}}$. The final manipulated graph is denoted as $\mathcal{G}' = (\mathcal{V}', \mathcal{E}')$.

**Unlearner's Perspective.** After training, model developers may observe that desired properties like fairness and robustness are compromised in the trained model $\mathcal{M}$, which can be modelled as lower accuracy on some data distributions. The objective, then, is to remove the influence of the manipulated training data $S_m$ on the affected distribution while maintaining performance on the remaining entities. By utilizing data monitoring strategies on a subset of the training data or using incorrect data detection tools like (Northcutt et al., 2021), it may be possible to identify a part of the manipulated entities $S_f \subseteq S_m$. For unlearning to be feasible, $S_f$ must be a representative subset of $S_m$. We only assume the type of affected entity (edges or nodes) is known to the model developer but do not assume any knowledge about the nature of manipulation. An unlearning method $U(\mathcal{M}, S_f, \mathcal{G}')$ is then used to mitigate the adverse effects of $S_m$, ideally by improving the accuracy on unseen samples from the affected distribution. An effective unlearning method should remove the impact of certain training data samples without degrading performance on the rest of the data or incurring the cost of retraining from scratch. Moreover, Retrain was previously considered a gold standard in privacy-oriented unlearning and graph unlearning, but Goel et al. (2024) showed that when the whole manipulated set is not known, retraining on the remaining data can reinforce the manipulation, implying it's not a gold standard for corrective unlearning.

**Metrics.** To evaluate the performance of unlearning methods in this setting, we use the metrics proposed by Goel et al. (2024):

1. **$\text{Acc}_{\text{aff}}$ :** It measures the clean-label accuracy of test set samples from the affected distribution. This metric captures the method's ability to *correct* the influence of the manipulated entities on unseen data through unlearning. As the affected distribution differs for each manipulation, we specify it when describing each evaluation.

2. **$\text{Acc}_{\text{rem}}$ :** It is defined as the accuracy of the remaining entities. This metric measures whether the unlearning maintains model performance on clean entities.

The metrics $\text{Acc}_{\text{aff}}$ and $\text{Acc}_{\text{rem}}$ were termed "Corrected Accuracy" ($\text{Acc}_{\text{corr}}$) and "Retain Accuracy" ($\text{Acc}_{\text{retain}}$) respectively by Goel et al. (2024). We chose alternative names to explicitly state which data distribution accuracy is measured on. In Section 5.1 we further specify what "the affected distribution" and "remaining entities" are for the different evaluation types we study.

**Goal.** An ideal corrective unlearning method should have high $\text{Acc}_{\text{aff}}$ even when a small fraction of $S_m$ is identified for deletion ($S_f$) while maintaining $\text{Acc}_{\text{rem}}$ and taking less computation time.

# 4 OUR METHOD: *Cognac*

Our proposed unlearning method, *Cognac*, requires access to the underlying graph $\mathcal{G}'$, the known set of entities to be deleted $S_f$, and the original model $\mathcal{M}$. We define $\mathcal{V}_f$ as the set of nodes whose influence is to be removed. For node unlearning, this is the same as the deletion entities $S_f$; for edge unlearning, this is the set of vertices connected to the edge set to be deleted. Manipulated data has two main adverse effects on the trained GNN: 1) Message passing can propagate the influence of the manipulated entities $S_m$ on their neighborhood, and 2) The layers learn transformations to fit potentially wrong labels in $S_m$. We tackle these two problems using separate components.

## 4.1 REMOVING ADVERSE EFFECTS ON NEIGHBORING NODES WITH COGN

The first question we address is: *How can we remove the influence of manipulated entities on their neighboring nodes?* This requires us first to identify the nodes affected by the manipulations and then mitigate the influence on their representations. Identifying affected nodes is challenging, as the impact of message passing from manipulated entities $S_m$ depends on the interference from messages of other neighboring nodes. Therefore, we use an empirical estimation to identify the affected nodes from each entity in the deletion set. On these nodes, we then perform contrastive unlearning, simultaneously pushing the representations of the affected nodes away from nodes in $\mathcal{V}_f$ while keeping them close to other nodes in their neighborhood. We call this component *Contrastive unlearning on Graph Neighborhoods* (**CoGN**), formalized below.

**Affected Node Identification.** To speed up our method, we make use of the fact that not all nodes in the $n$-hop neighbourhood of the manipulated nodes may be affected enough by the attack. To find the most affected nodes, we invert the features of $v \in V_f$ and select neighbouring nodes where final output logits are changed the most. Formally, the inversion is performed by the transformation $\vec{1} - \mathcal{X}_v \forall v \in \mathcal{V}_f$, leading to a new feature matrix $\chi'$, where $\mathcal{X}_v$ represents a one-hot-encoding vector. We then compute the difference in the original output logits $\mathcal{M}(\chi)$, and those obtained by on the new feature matrix, $\mathcal{M}(\chi')$ given by: $\Delta\chi = |\mathcal{M}(\chi') - \mathcal{M}(\chi)|$. The top $k\%$ nodes with the most change $\Delta\chi$ are selected as the affected set of entities, from which we remove the influence of the manipulation using *CoGN*. More details and ablations in Appendix:I.

**Contrastive Unlearning.** To remove the influence of the deletion set $S_f$ on the affected nodes identified in the previous step, we can optimize a loss function that updates the weights such that the final layer logits of $S_f$ and the affected nodes are pushed away. However, this alone will lead to unrestricted separation and damage the quality of learned representations. To prevent this, we also counterbalance the loss with another term that penalizes moving away from logits of neighboring nodes not in the deletion set $S_f$. For each node $v \in \mathcal{S}$, let $z_v$ represent its internal embedding, with $p \in \mathcal{N}(v)$ and $n \in \mathcal{V}_f$ serving as the positive and negative samples, respectively. We use the following unsupervised contrastive loss:

$$\mathcal{L}_c = -\log(\sigma(z_v^\top z_p)) - \log(\sigma(-z_v^\top z_n)) \tag{1}$$

The loss is similar to the one used in GraphSAGE (Hamilton et al., 2017), but only updates affected nodes to make their representations dissimilar from the deletion set while keeping them similar to the remaining nodes in their neighborhood. We choose an unsupervised loss function to fix representations even in mislabeling, which is essential when the manipulated set is not fully known ($S_f \subset S_m$).

## 4.2 Unlearning old labels with AC↯DC

Next, we ask: *Can we undo the effect of the task loss $\mathcal{L}_{\text{task}}$ explicitly learning to fit the node representations of $S_m$ to potentially wrong labels?* We do this by performing gradient ascent on $S_f$, which non-directionally maximizes the training loss concerning the old labels. Ascent alone aggressively leads to arbitrary forgetting of useful information, so we counterbalance it by alternating with steps that minimize the task loss on the remaining data. More precisely, we perform gradient ascent on $\mathcal{V}_f$ and gradient descent on $\mathcal{V}_r$, iteratively on the original GNN $\mathcal{M}$.

$$\mathcal{L}_a = -\mathcal{L}_{\text{task}}(\mathcal{V}_f), \ \mathcal{L}_d = \mathcal{L}_{\text{task}}(\mathcal{V}_r) \tag{2}$$

While variants of ascent on $S_f$ and descent on remaining data have been studied for image classification (Kurmanji et al., 2023) and language models (Yao et al., 2023), we find the need for a specific optimization strategy to achieve corrective unlearning on graphs. The challenge arises when $S_f \subset S_m$, as the remaining data could still contain manipulated entities, which we aim to avoid reinforcing. However, in realistic scenarios, the manipulated entities $S_m$ typically constitute a small fraction of the training data, allowing us to mitigate their impact through ascent on the representative subset $S_f$.

This requires a careful balance between ascent and descent, which we can achieve by using two different optimizers and starting learning rates for these steps. This insight is similar to prior work in Generative Adversarial Networks (GANs) (Heusel et al., 2017). The starting learning rates for both ascent and descent are hyperparameters to be tuned, and usually, we find a lower learning rate for ascent leads to better results. The importance of decoupling optimizers is shown by results in Table 3. Thus, we call this component *Ascent Descent de↯coupled* (*AC↯DC*) to emphasize the distinction from existing variants of ascent on $S_f$ and descent on remaining data.

For our final method ***Cognac***, we alternate steps of *CoGN*, which fixes representations of affected neighborhood nodes, and *AC↯DC*, which unlearns potentially wrong labels introduced by $S_m$. The complete algorithm is formally detailed in the Appendix B.

## 5 EXPERIMENTAL SETUP

### 5.1 EVALUATIONS

Given a fixed budget of samples to manipulate, ideal corrective unlearning evaluations should maximally deteriorate model performance on the affected distribution so that there is a wide gap between clean and poisoned model performance to measure unlearning method progress. We thus evaluate using attacks not constrained on stealthiness. Lingam et al. (2024) show that binary label flip manipulation attacks, where a fraction of labels are swapped between two chosen classes, are stronger than multi-class manipulations, theoretically and empirically, on GNNs. Building on this, we use two targeted attacks to evaluate corrective unlearning on graph data.

**Spurious Edge Addition.** We first describe a graph-specific manipulation where an attacker can add edges to the graph topology (Bojchevski & Günnemann, 2019a). Such an attack can occur in real-world settings like social networks, where attackers can create fake accounts and follow targeted accounts, strengthening their connection in the underlying graph. Similarly, attackers could manipulate knowledge graphs by adding links between unrelated concepts (Xi et al., 2023; Zhang et al., 2019; Zhao et al., 2024), or manipulate search engine results by adding fake cross-references (Gyongyi & Garcia-Molina, 2005). Prior GNN unlearning work (Wu et al., 2023a; Li et al., 2024b) has also evaluated adversarial edge attacks but in an untargeted setting, making the evaluation weak. In our formulation, the attacker selects two classes, samples random pairs of nodes uniformly, with one from each class, and adds edges between them. This targets the underlying homophily assumption in message passing, leading to representations of the two classes being entangled when training the model. Hence, this attack reduces the model's accuracy on samples from the two classes, which form the affected distribution. Thus, the unlearning goal is to improve accuracy on the test set samples of the two targeted classes ($\text{Acc}_{\text{aff}}$). For $\text{Acc}_{\text{rem}}$, we measure accuracy on test set samples from the remaining classes.

**Label Manipulation.** Next, we study a label-only manipulation that models settings where model developers source external annotations on their data. We focus on systematic mislabeling, which can occur in an adversarial context where an attacker wants the model to confuse two classes due to annotator biases or misinterpretations of potentially ambiguous guidelines. We use the Interclass Confusion (IC) Test (Goel et al., 2022), where the attacker picks two classes again, swapping the labels between nodes from the two classes. This attack also entangles the representations of the two classes, reducing the model's accuracy on them, which forms the affected distribution. Once again, the unlearning goal is to improve accuracy on the test set samples of the two targeted classes ($\text{Acc}_{\text{aff}}$). For $\text{Acc}_{\text{rem}}$, we measure accuracy on test set samples from the remaining classes.

### 5.2 BASELINES

We evaluate four popular graph unlearning methods and adapt one popular i.i.d unlearning method for graphs. For reference, we also report results for the Original model, Retrain which trains a new model without $S_f$, and an Oracle trained on the whole training set without manipulations, indicating an upper bound on what can be achieved. The Oracle has correct labels for the unlearning entities, information that the unlearning methods cannot access.

**Existing Unlearning Methods.** We choose five methods as baselines where unlearning incorrect data explicitly motivates the method. (1) **GNNDelete** (Cheng et al., 2023) adds a deletion operator after each GNN layer and trains them using a loss function to randomize the prediction probabilities of deleted edges while preserving their local neighborhood representation, keeping the original GNN weights unchanged. (2) **GIF** (Wu et al., 2023a) draw from a closed-form solution for linear-GNN to measure the structural influence of deleted entities on their neighbors. Then, they provide estimated GNN parameter changes for unlearning using the inverse Hessian of the loss function. (3) **MEGU** (Li et al., 2024b) finds the highly influenced neighborhood (HIN) of the unlearning entities and removes their influence over the HIN while maintaining predictive performance and forgetting the deletion set using a combination of losses. (4) **UtU** (Tan et al., 2024) proposes *zero-cost* edge-unlearning by removing the edges to be deleted during inference for blocking message propagation from nodes linked to these edges. Finally, we include a popular unlearning method studied in i.i.d classification settings. (5) **SCRUB** (Kurmanji et al., 2023) employs a teacher-student framework with alternate steps of distillation away from the forget set and towards the retain set. For edge

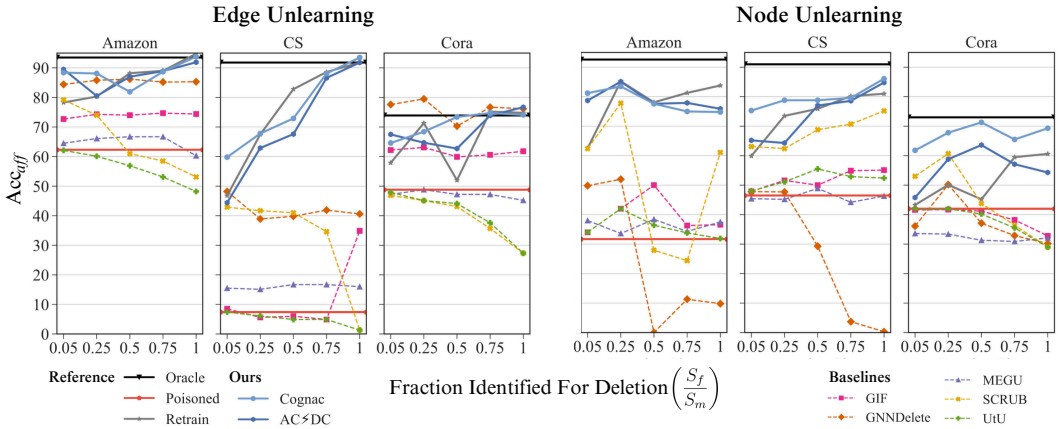

Figure 2: **Corrective Unlearning Results.** We report the accuracy on the affected classes $\text{Acc}_{\text{aff}}$ across different fractions of the manipulation set known for deletion ($S_f/S_m$). Prior methods perform poorly, except for GNNDelete, which does achieve unlearning in some settings. AC⚡DC, despite not being graph-specific, performs much better. *Cognac* adds graph awareness to it, improving results on CS and Cora, unlearning the effect of the manipulation with just 5% of the manipulation set known.

unlearning, we use SCRUB by taking nodes across spuriously added edges as the forget set and the rest as the retain set.

## 5.3 BENCHMARKING DETAILS

We now describe design choices made for benchmarking, first specifying the datasets and architectures, then how to ensure a fair comparison between methods.

**Models and Datasets.** We report results using Graph Convolutional Networks (GCN) (Kipf & Welling, 2017) and also show the same trends hold for Graph Attention Transformers (GAT) (Veličković et al., 2018) in Appendix D. We evaluate the methods on three benchmark datasets: CoraFull (Cora) (Bojchevski & Günnemann, 2017), Coauthor CS (CS) (Shchur et al., 2019), and Amazon Photos (Amazon) (McAuley et al., 2015). Data set size details and the corresponding classes and number of entities manipulated are provided in Table 4 in Appendix A.

Ensuring a fair comparison of unlearning methods can be tricky as there are multiple desiderata: unlearning, maintaining utility, and computational efficiency, and hyperparameter tuning of the methods can particularly affect results on GNNs. Next, we describe our efforts towards this.

**Unlearning Time.** To simplify comparisons to just two axes, $\text{Acc}_{\text{aff}}$, and $\text{Acc}_{\text{rem}}$, we fix a maximum cutoff of time an unlearning method can take, as motivated by Maini et al. (2024). We chose this cutoff as 25% of the original model training time. We pick the best model checkpoint during training for each method, which could be achieved earlier than this (Reported in Appendix E. All experiments were run on a machine with Intel Xeon CPUs and two dedicated RTX 5000 GPUs.

**Hyperparameter Tuning.** We perform extensive hyperparameter tuning for all unlearning methods using Optuna (Akiba et al., 2019) with a TPESampler (Tree-structured Parzen Estimator) Algorithm. We ensure the hyperparameter ranges searched include any values specified in the original papers that proposed the methods. The optimization target is an average of $\text{Acc}_{\text{aff}}$ and $\text{Acc}_{\text{rem}}$, computed on the validation set. We report averaged results across five seeds. Method-specific hyperparameter ranges and scatter plots across hyperparameters for each method are provided in Appendix C.

Table 1: **Accuracy on remaining distribution for datasets and attacks.** We average accuracy on the remaining classes $Acc_{rem}$ across the deletion set sizes tested, and report the difference from Original as ideal methods should be similar or better. We find prior methods, especially GNNDelete and GIF, lead to large drops in accuracy, which is much milder for our methods.

| Method | Amazon | | CS | | Cora | |
|---|---|---|---|---|---|---|
| | **Node** | **Edge** | **Node** | **Edge** | **Node** | **Edge** |
| Original | $94.0_{\pm 0.0}$ | $94.9_{\pm 0.0}$ | $90.4_{\pm 0.0}$ | $89.9_{\pm 0.0}$ | $56.0_{\pm 0.0}$ | $55.4_{\pm 0.0}$ |
| *Cognac* | $-0.7_{\pm 0.5}$ | $-3.9_{\pm 0.0}$ | $-0.8_{\pm 0.4}$ | $-1.3_{\pm 0.1}$ | $0.0_{\pm 0.4}$ | $-1.1_{\pm 1.3}$ |
| CoGN | $-0.7_{\pm 0.6}$ | $-2.7_{\pm 0.0}$ | $-2.2_{\pm 1.7}$ | $-0.5_{\pm 0.2}$ | $0.8_{\pm 0.3}$ | $-1.0_{\pm 1.2}$ |
| AC$\natural$DC | $-0.5_{\pm 0.3}$ | $-2.7_{\pm 0.0}$ | $-1.1_{\pm 0.3}$ | $-0.9_{\pm 0.0}$ | $1.5_{\pm 0.0}$ | $-1.4_{\pm 0.0}$ |
| GNNDelete | $-24.7_{\pm 6.1}$ | $-2.7_{\pm 0.5}$ | $-5.3_{\pm 1.1}$ | $-5.0_{\pm 1.1}$ | $-6.1_{\pm 1.4}$ | $-11.5_{\pm 0.1}$ |
| GIF | $-18.5_{\pm 0.1}$ | $-3.3_{\pm 0.1}$ | $-18.5_{\pm 0.5}$ | $-3.7_{\pm 1.7}$ | $-1.0_{\pm 0.4}$ | $-8.8_{\pm 0.7}$ |
| MEGU | $-8.5_{\pm 11.5}$ | $-1.1_{\pm 1.2}$ | $-1.7_{\pm 0.2}$ | $-1.5_{\pm 2.5}$ | $-6.5_{\pm 1.0}$ | $-1.0_{\pm 0.8}$ |
| UtU | $0.5_{\pm 0.1}$ | $0.0_{\pm 0.0}$ | $0.0_{\pm 0.1}$ | $0.0_{\pm 0.0}$ | $0.0_{\pm 0.0}$ | $0.0_{\pm 0.0}$ |
| SCRUB | $-9.4_{\pm 3.7}$ | $-0.8_{\pm 0.0}$ | $0.0_{\pm 0.2}$ | $-0.6_{\pm 0.0}$ | $-0.7_{\pm 0.2}$ | $0.3_{\pm 0.0}$ |

## 6 RESULTS & DISCUSSION

We now report our results, first showing comparisons of our method to existing methods across the manipulation types and datasets, followed by ablations of our method and analysis of what can be achieved in this setting.

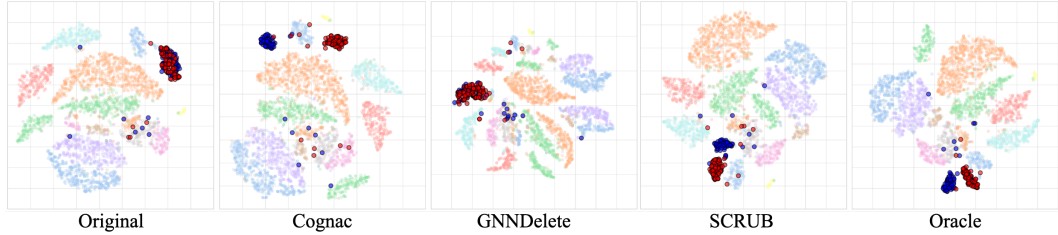

Figure 3: **Pre-final layer embeddings for test nodes of the CS Dataset for the node unlearning of class confusion**. *Cognac* can resolve the class confusion: The affected distribution embeddings (highlighted by red and blue) are fully entangled in the original trained model while after unlearning with *Cognac* the embeddings are well separated and clustered. SCRUB manages to do it to a certain extent, and GNNDelete cannot recover from it.

### 6.1 MAIN RESULTS

Figure 2 contain unlearning measurements upon varying the fraction of the manipulation set known for unlearning $(S_f/S_m)$. Table 1 accompanies this with utility measurements averaged across the deletion fractions for each method. We make three main observations:

**1. Existing unlearning methods perform poorly even when $|S_f| = |S_m|$.** First, observing the rightmost points in Figure 2, we can see that existing methods fail to unlearn across manipulation types even when the whole manipulation set is known. The only exception is GNNDelete on the edge unlearning task, but with up to $24.7\%$ drops in $Acc_{rem}$ as seen in Table 1, which has also been observed before as *Overforgetting* in Tan et al. (2024). UtU fails to unlearn the effects of either of the attacks as simply unlinking on the forward pass does not sufficiently counteract the influence on neighbors and weights. Both SCRUB and MEGU use a KL Loss to keep predictions on the remaining data close to the original model, which could be detrimental when done on unidentified manipulation set entities and other affected neighbors.

**2. AC$\natural$DC shows strong results but improves with CoGN.** AC$\natural$DC, a method with no graph-specific aspects, performs quite strongly, beating all the methods we compare to without losing utility on the remaining classes. MEGU and GIF were evaluated on removing adversarial edges, and GNNDelete also mentions incorrect data as one of its key applications. Yet, despite extensive hyperparameter search, they are beaten by a method with no special graph components. This highlights the importance of strong evaluations for graph unlearning, a bar our evaluations cross as they at least demonstrate the failure of existing methods. These findings raise the question: are graph-specific unlearning methods even needed? We find yes. *Cognac*, which adds the graph-aware CoGN step to AC$\natural$DC dominates AC$\natural$DC on CS and Cora, the more complex datasets, while performing similarly on Amazon.

**3. *Cognac* performs strongly even at $5\%$ of $\mathbf{S_m}$ known.** We observe that *Cognac* performs the best across all the datasets and manipulation types, recovering most of the accuracy on the affected distribution even when $5\%$ of the manipulated set is known. We even outperform Retrain in the realistic settings when $S_m$ is not fully known, as we utilize negative information, i.e., push influenced neighbors away from the identified deletion set in CoGN and perform gradient ascent on old labels in AC$\natural$DC.

Overall, these results demonstrate the efficacy of *Cognac* in removing different types of manipulations at a tiny fraction ($5\%$) of the manipulation set $S_m$ known, which shows significant progress on the challenge of Corrective Unlearning in th graph setting. We also visualize each method's intermediate GNN layer embeddings after unlearning in Figure 3 where we can clearly notice *Cognac* can remove the class confusion effect, fixing the model's internal representations.

### 6.2 Why $\text{Acc}_{\text{aff}}$ sometimes reduces as identified manipulated entities increase?

We find an interesting trend that sometimes, as more of the manipulation set $(S_m)$ is known and used as the deletion set $(S_f)$ (going left to right in Figure 2), $\text{Acc}_{\text{aff}}$ reduces. This can seem counter-intuitive, as one would expect the accuracy of affected classes to improve as more samples are used for unlearning. We hypothesize unlearning a larger fraction of the manipulation set reduces $\text{Acc}_{\text{aff}}$ due to two factors that adversely affect the neighborhoods of the nodes removed, which typically have other nodes of the affected classes due to homophily. First, in the case of label manipulation, when we model it as node unlearning for consistency with prior work, we lose correct information about the graph structure. Second, when modifying the graph structure, i.e., removing some edges or nodes changes the feature distribution of their neighboring nodes after the message passes, making it out of distribution for the learned GNN layers. The same rationale is why the test nodes are kept in the graph structure (without optimizing the task loss for them) during training (Kipf & Welling, 2017). We investigate this by adding an ablation where in the unlearning of the label manipulation, instead of unlearning the whole node, we keep the structure, i.e., the node and connected edges, but unlearn the features and labels.

Table 2: **Ablating unlearning performance on label manipulation with and without unlinking.** We report the accuracy on the affected classes $\text{Acc}_{\text{aff}}$ for unlearning the label manipulation on Cora both when the full and a subset (25%) of the manipulated set is used for deletion. We find not removing the structural information leads to a significant improvement in $\text{Acc}_{\text{aff}}$, especially when more entities are deleted. This also demonstrates that the unlearner can obtain better models if specific information about what is manipulated is available.

| Method | 0.25 | | 1 | |
|---|---|---|---|---|
| | **Linked** | **Unlinked** | **Linked** | **Unlinked** |
| Oracle | $73.0_{\pm 0.0}$ | $73.0_{\pm 0.0}$ | $73.0_{\pm 0.0}$ | $73.0_{\pm 0.0}$ |
| Original | $42.0_{\pm 0.0}$ | $42.0_{\pm 0.0}$ | $42.0_{\pm 0.0}$ | $42.0_{\pm 0.0}$ |
| *Cognac* | $\mathbf{64.8_{\pm 0.9}}$ | $\mathbf{67.8_{\pm 3.2}}$ | $\mathbf{77.2_{\pm 1.0}}$ | $\mathbf{69.3_{\pm 1.3}}$ |
| CoGN | $64.6_{\pm 0.0}$ | $63.9_{\pm 2.2}$ | $69.3_{\pm 0.0}$ | $62.3_{\pm 1.1}$ |
| GNNDelete | $35.2_{\pm 2.5}$ | $50.2_{\pm 1.9}$ | $21.9_{\pm 4.5}$ | $30.2_{\pm 5.3}$ |
| MEGU | $40.8_{\pm 1.6}$ | $33.4_{\pm 0.4}$ | $41.2_{\pm 1.5}$ | $32.1_{\pm 1.3}$ |
| SCRUB | $45.7_{\pm 0.0}$ | $60.7_{\pm 0.0}$ | $41.1_{\pm 0.0}$ | $29.0_{\pm 0.0}$ |
| AC$\natural$DC | $61.7_{\pm 0.0}$ | $58.8_{\pm 0.0}$ | $63.7_{\pm 0.0}$ | $54.3_{\pm 0.0}$ |

As observed in Table 2, retaining the node structure leads to large improvements in $\text{Acc}_{\text{aff}}$ when the deletion set is larger (the full set of manipulated entities), while not benefiting much when the deletion set is smaller. In the full manipulation set deletion setting, *Cognac* even slightly outperforms Oracle. This highlights how unlike traditional node unlearning in graphs, removing the nodes is not always the best way to unlearn manipulations. They can simply be moved from the train set to the test set to still partake in message passing, so the task loss is not optimized over wrong labels.

### 6.3 Importance of Decoupled Optimizers for Ascent and Descent

While ascent-descent has been studied in prior unlearning work, it often performs poorly due to unstable loss dynamics. We find this can be fixed with a simple trick: using two learning rates and different instances of Adam instead of coupling the optimization of both steps. Adding another hyperparameter for the ascent learning rate is necessary as the ascent is not always needed when the original training data labels are correct. For example, we do find that the ascent learning rate is set to nearly zero during automatic hyperparameter selection for our edge unlearning evaluation, where labels are not manipulated. This leaves the question of how much two different instances of the same optimizer help. We thus test two ablations: Using a single optimizer with two learning rates and also using a combined loss function for ascent and descent instead of alternating steps. In Table 3 we report results on 25% and 100% of the manipulated nodes identified for Cora. We find AC♮DC leads to almost 20% better $\text{Acc}_{\text{aff}}$ than using a single optimizer with alternating ascent descent, and $5-10\%$ better $\text{Acc}_{\text{aff}}$ than using a combined loss function. This justifies our contribution of decoupling optimizers in AC♮DC.

Table 3: **Ablations of AC♮DC at** $(S_f/S_m)$ **= 0.25 and 1.00 on Cora node unlearning**. We ablate the AC♮DC part of our method, even when the losses are combined in a single loss term.

| Method | 0.25 | | 1.00 | |
|---|---|---|---|---|
| | $\text{Acc}_{\text{aff}}$ | $\text{Acc}_{\text{rem}}$ | $\text{Acc}_{\text{aff}}$ | $\text{Acc}_{\text{rem}}$ |
| Single Optimizer 1 LR Alternating Ascent Descent | 38.3 | 56.0 | 29.0 | 56.2 |
| Single Optimizer 2 LRs Combined Loss | 47.5 | 56.7 | 49.6 | 55.1 |
| AC♮DC | 58.8 | 57.4 | 54.3 | 56.1 |

## 7 Limitations and Conclusion

In this work, we study the corrective unlearning problem for GNNs, where model developers try to remove the adverse effects of manipulated training data from a trained GNN, with realistically only a fraction of it identified for deletion. Our work relies on the homophily assumption, not catering to heterophilic graphs (Wang et al., 2024). Our evaluations could have been stronger, and may not match real-world complexities, where multiple manipulations can occur simultaneously, and attackers have constraints such as avoiding detection. Successful unlearning on our evaluations does not guarantee arbitrary real-world unlearning, especially against adaptive attacks.

Still, our evaluations are sufficient to show that existing unlearning methods perform poorly at removing adverse effects of manipulations from GNNs, even in the unrealistic setting of the full manipulation set being known. We propose a new method *Cognac* that achieves two crucial effects necessary for corrective unlearning. First, *Cognac* identifies and fixes representations of neighborhood nodes affected by the deletion set using contrastive finetuning. Second, *Cognac* moves away from potentially wrong deletion set labels using gradient ascent, stabilized by continuing optimization of the task loss on the remaining data with a decoupled optimizer. While our method does not provide any theoretical guarantees, to the best of our knowledge, *Cognac* is the first method to unlearn class confusion manipulations with access to as little as 5% of the manipulated data, recovering most of the accuracy on the affected distribution. With additional information, it nearly matches a strong oracle with full correct training data. We hope this sparks interesting future work on developing stronger evaluations and theoretical understanding for graph corrective unlearning in the GNN Robustness and Machine Unlearning community.

## 8 REPRODUCIBILITY STATEMENT

We provide our entire codebase in the supplementary materials to ensure reproducibility, along with the optimal hyperparameters for each configuration. All artefacts will be publicly released, complete with comprehensive instructions for their use, including scripts and code for generating plots.

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

## A  DATASET DETAILS

In Table 4, we discuss the number of nodes, edges, and classes in the Cora, Amazon, and CS datasets. We have used Cora Full, which is significantly larger and has higher class diversity than the Cora dataset commonly referenced in GNN literature (which has 2708 nodes, 5429 edges, and 7 classes).

Table 4: **Dataset and manipulation statistics**. The number of nodes and edges reported here is for the whole dataset. From this, we use a 60/20/20 split for train/validation/test. The last two columns show the percentage of training data manipulated in the node unlearning and edge unlearning evaluation, respectively. The percentage of nodes and edges added are relative to the existing number of training nodes and the number of edges in the graph, respectively.

| Dataset | # Classes | # Nodes | # Edges | Nodes Manipulated (%) | Edges Added (%) |
|---------|-----------|---------|---------|-----------------------|-----------------|
| Cora | 70 | 18,800 | 125,370 | 1.50 | 0.60 |
| CS | 15 | 18,333 | 163,788 | 2.00 | 2.00 |
| Amazon | 8 | 7,487 | 238,086 | 12.00 | 4.20 |

# B  FORMAL DESCRIPTION OF *Cognac*

In this section, we outline the formal procedure of our proposed unlearning method, *Cognac* designed to effectively remove the influence of manipulated data from GNNs. First, the algorithm identifies the nodes affected by the manipulation and then applies a contrastive learning-based approach to unlearn their influence. The key steps include identifying the influenced nodes, performing contrastive learning to re-optimize the embeddings, and minimizing classification loss on the unaffected nodes while maximizing it on the manipulated set. The complete algorithm is detailed in Algorithm 1.

---

**Algorithm 1** COGNAC

---

**Require:** GNN $M$, Graph $G = (V, E, X)$, Deletion set $S_f$, Hyperparameters $\Theta$
**Ensure:** Unlearned GNN $M^*$

1: $S \leftarrow$ IDENTIFYAFFECTEDNODES$(M, X, S_f, E, \Theta)$
2: $P \leftarrow$ SAMPLEPOSITIVES$(S, E, S_f)$
3: $N \leftarrow$ SAMPLENEGATIVES$(S, S_f)$
  *// Overall unlearning process*
4: **for** step $= 1$ **to** $\Theta$.num_steps **do**
  *// contrastive unlearning phase*
5:   **for** epoch $= 1$ **to** $\Theta$.contrast_epochs **do**
6:     $Z \leftarrow M(X)$
7:     $L_c \leftarrow \sum_{v \in S} \left( -\log(\sigma(Z_v^\top Z_{P_v})) - \log(\sigma(-Z_v^\top Z_{N_v})) \right)$
8:     $M \leftarrow$ OPTIMIZE$(M, L_c)$
9:   **end for**
  *// gradient ascent on $S_f$, and gradient descent on $V \setminus S_f$*
10:   **for** epoch $= 1$ **to** $\Theta$.ascent_descent_epochs **do**
11:     $L_a \leftarrow -$CROSSENTROPY$(M(X)_{S_f}, Y_{S_f})$
12:     $M \leftarrow$ OPTIMIZE$(M, L_a)$
13:     $L_d \leftarrow$ CROSSENTROPY$(M(X)_{V \setminus S_f}, Y_{V \setminus S_f})$
14:     $M \leftarrow$ OPTIMIZE$(M, L_d)$
15:   **end for**
16: **end for**

17: **return** $M$

18: **function** IDENTIFYAFFECTEDNODES$(M, X, S_f, E, \Theta)$
19:   $X' \leftarrow$ INVERTFEATURES$(X, S_f, E)$
20:   $\Delta \leftarrow |M(X') - M(X)|$
21:   **return** TOPK$(\Delta, \Theta.k)$
22: **end function**

---

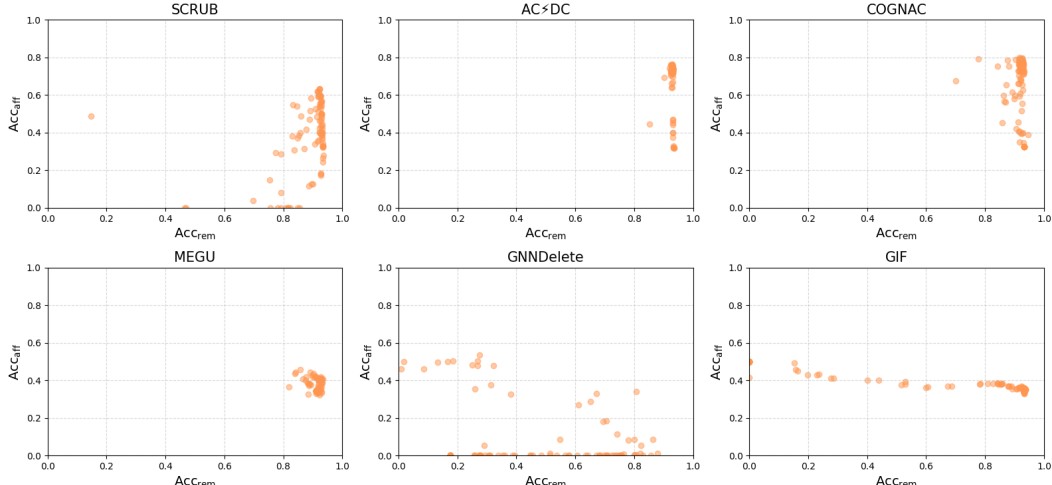

Figure 4: **Hyperparameter runs for** $S_f = 1.0$ **on Amazon**. Scores of various hyperparameter trial runs. The best hyperparameters are selected according to the run achieving the best value for the average of $\text{Acc}_{\text{rem}}$ and $\text{Acc}_{\text{aff}}$.

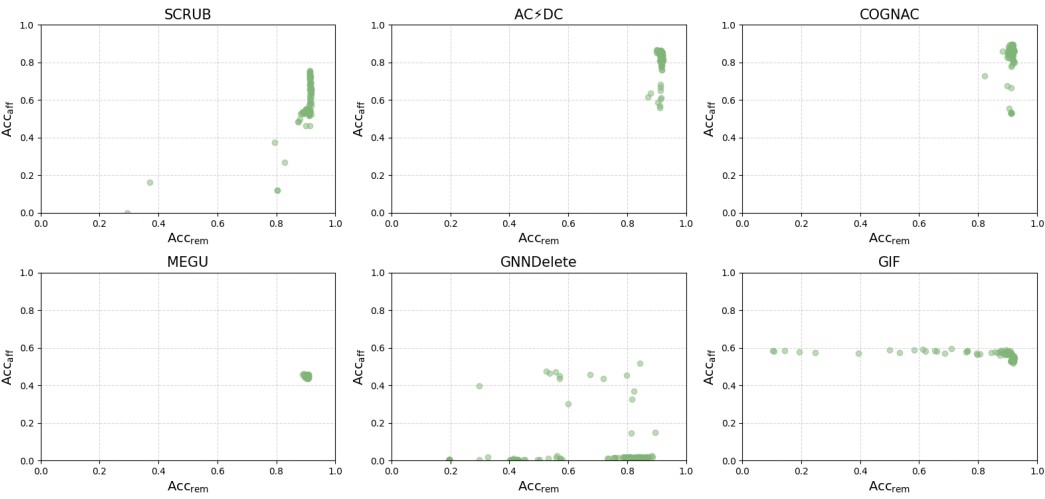

Figure 5: **Hyperparameter runs for** $S_f = 1.0$ **on CS**. Scores of various hyperparameter trial runs. The best hyperparameters are selected according to the run achieving the best value for the average of $\text{Acc}_{\text{rem}}$ and $\text{Acc}_{\text{aff}}$.

## C  HYPERPARAMETER TUNING

We perform hyperparameter tuning for each combination for attack, dataset, unlearning method, and identified fraction of deletion set ($S_f$). The optimization target is an average of $\text{Acc}_{\text{aff}}$ and $\text{Acc}_{\text{rem}}$, computed on the validation set. For each setting, we run 100 trials with hyperparameters selected using the TPESampler (Tree-structured Parzen Estimator) algorithm. In Figure 6, we report $\text{Acc}_{\text{aff}}$ and $\text{Acc}_{\text{rem}}$ scores for each hyperparameter tuning trial. Across hyperparameter runs, existing graph-based unlearning methods, barring MEGU, vary drastically across different sets of hyperparameters. On the other hand, our proposed method *Cognac* and its ablations show consistently high scores across hyperparameters, showcasing *Cognac*'s robustness to hyperparameter tuning.

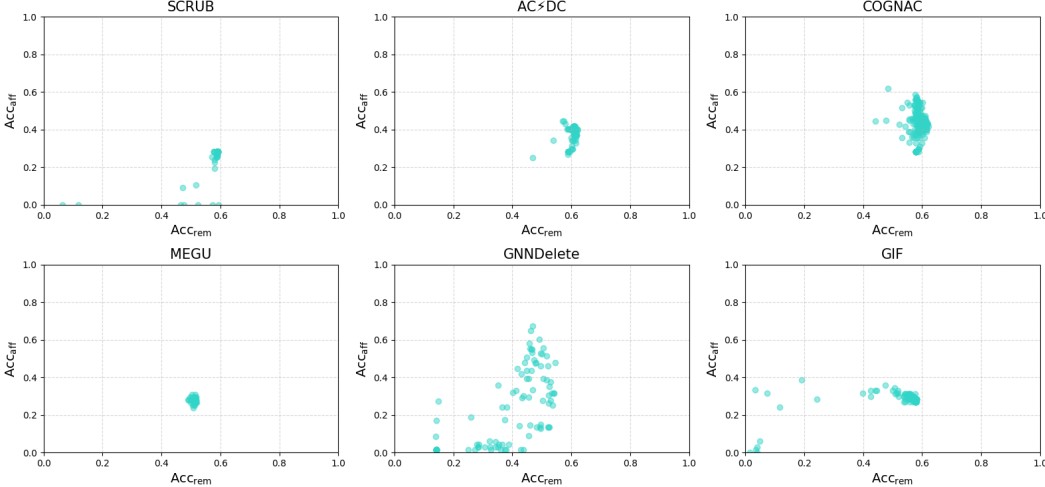

Figure 6: **Hyperparameter runs for** $S_f = 1.0$ **on Cora**. Scores of various hyperparameter trial runs. The best hyperparameters are selected according to the run achieving the best value for the average of $\text{Acc}_{\text{rem}}$ and $\text{Acc}_{\text{aff}}$.

## D  RESULTS ON ANOTHER ARCHITECTURE: GRAPH ATTENTION NETWORKS

To provide a comprehensive comparison between *Cognac* and other methods, we provide results on commonly used GNN backbone architectures - GCN and GAT.

**Graph Convolutional Network (GCN)** is a method for semi-supervised classification of graph-structured data. It employs an efficient layer-wise propagation rule derived from a first-order approximation of spectral convolutions on graphs.

**Graph Attention Network (GAT)** employs computationally efficient masked self-attention layers that assign varying importance to neighborhood nodes without needing the complete graph structure upfront, thereby overcoming many theoretical limitations of earlier spectral-based methods.

Table 5: **Comparison of** $\text{Acc}_{\text{aff}}$ **for Node Unlearning on GAT backbone.** The GAT model trained on Cora for different values of $(S_f/S_m)$

| Method | 0.05 | 0.25 | 0.50 | 0.75 | 1.00 |
|---|---|---|---|---|---|
| Oracle | $69.2_{\pm 0.0}$ | $69.2_{\pm 0.0}$ | $69.2_{\pm 0.0}$ | $69.2_{\pm 0.0}$ | $69.2_{\pm 0.0}$ |
| Retrain | $\mathbf{44.7}_{\pm \mathbf{4.6}}$ | $47.3_{\pm 6.4}$ | $\mathbf{52.4}_{\pm \mathbf{5.6}}$ | $47.0_{\pm 4.1}$ | $51.1_{\pm 4.3}$ |
| Original | $39.3_{\pm 0.0}$ | $39.3_{\pm 0.0}$ | $39.3_{\pm 0.0}$ | $39.3_{\pm 0.0}$ | $39.3_{\pm 0.0}$ |
| *Cognac* | $39.1_{\pm 1.1}$ | $\mathbf{49.8}_{\pm \mathbf{3.7}}$ | $51.5_{\pm 2.6}$ | $\mathbf{47.1}_{\pm \mathbf{1.7}}$ | $\mathbf{53.9}_{\pm \mathbf{3.3}}$ |
| AC$\notin$DC | $39.3_{\pm 0.0}$ | $46.0_{\pm 2.6}$ | $44.2_{\pm 3.0}$ | $46.4_{\pm 1.8}$ | $50.2_{\pm 3.1}$ |
| GNNDelete | $37.4_{\pm 0.6}$ | $42.1_{\pm 4.6}$ | $30.0_{\pm 4.4}$ | $17.3_{\pm 7.4}$ | $41.5_{\pm 0.0}$ |
| GIF | $38.3_{\pm 0.0}$ | $37.6_{\pm 0.7}$ | $35.9_{\pm 0.5}$ | $33.7_{\pm 0.0}$ | $32.8_{\pm 0.0}$ |
| MEGU | $30.1_{\pm 2.3}$ | $29.9_{\pm 2.1}$ | $31.2_{\pm 0.7}$ | $30.9_{\pm 2.2}$ | $31.2_{\pm 2.4}$ |
| UtU | $38.3_{\pm 0.0}$ | $33.6_{\pm 0.0}$ | $35.5_{\pm 0.0}$ | $31.8_{\pm 0.0}$ | $32.8_{\pm 0.0}$ |
| SCRUB | $39.3_{\pm 0.0}$ | $34.6_{\pm 0.0}$ | $36.5_{\pm 0.0}$ | $33.7_{\pm 0.0}$ | $33.7_{\pm 0.0}$ |

*Cognac* also performs competitively with a GAT backbone. When 5% of $S_m$ is known, *Cognac* beats SCRUB within the standard deviation. For higher fractions, we achieve greater $\text{Acc}_{\text{aff}}$ than the benchmark graph unlearning methods with large margins, often beating the performance of retraining the GNN from scratch. These results indicate that benchmark graph unlearning methods used for comparison cannot recover from the impact of the label flip poison. In contrast, our method is much closer to Oracle's performance. *Cognac* also maintains $\text{Acc}_{\text{rem}}$ across $(S_f/S_m)$ fractions.

Table 6: **Comparison of** $\text{Acc}_{\text{rem}}$ **for Node Unlearning on GAT backbone.** The GAT model trained on Cora for different values of $(S_f/S_m)$

| Method | 0.05 | 0.25 | 0.50 | 0.75 | 1.00 |
|---|---|---|---|---|---|
| Oracle | $52.7_{\pm 0.0}$ | $52.7_{\pm 0.0}$ | $52.7_{\pm 0.0}$ | $52.7_{\pm 0.0}$ | $52.7_{\pm 0.0}$ |
| Retrain | $51.8_{\pm 2.8}$ | $53.5_{\pm 2.3}$ | $54.1_{\pm 1.4}$ | $54.7_{\pm 1.4}$ | $54.8_{\pm 2.5}$ |
| Original | $52.5_{\pm 0.0}$ | $52.5_{\pm 0.0}$ | $52.5_{\pm 0.0}$ | $52.5_{\pm 0.0}$ | $52.5_{\pm 0.0}$ |
| *Cognac* | $52.8_{\pm 0.0}$ | $50.9_{\pm 2.0}$ | $50.8_{\pm 2.1}$ | $52.6_{\pm 0.5}$ | $52.7_{\pm 0.6}$ |
| AC$\natural$DC | $52.1_{\pm 0.0}$ | $51.5_{\pm 1.6}$ | $52.6_{\pm 0.3}$ | $52.0_{\pm 1.1}$ | $53.1_{\pm 0.9}$ |
| GNNDelete | $49.6_{\pm 0.4}$ | $45.9_{\pm 1.7}$ | $47.7_{\pm 1.3}$ | $44.5_{\pm 0.2}$ | $34.0_{\pm 0.0}$ |
| GIF | $52.4_{\pm 0.0}$ | $50.1_{\pm 0.4}$ | $51.7_{\pm 0.1}$ | $50.3_{\pm 0.1}$ | $51.6_{\pm 0.1}$ |
| MEGU | $41.9_{\pm 1.2}$ | $42.2_{\pm 0.9}$ | $42.5_{\pm 0.4}$ | $42.6_{\pm 0.5}$ | $42.7_{\pm 0.4}$ |
| UtU | $52.4_{\pm 0.0}$ | $52.4_{\pm 0.0}$ | $52.2_{\pm 0.0}$ | $52.2_{\pm 0.0}$ | $52.2_{\pm 0.0}$ |
| SCRUB | $52.1_{\pm 0.0}$ | $52.4_{\pm 0.0}$ | $52.4_{\pm 0.0}$ | $52.3_{\pm 0.0}$ | $52.5_{\pm 0.0}$ |

# E UNLEARNING TIMES

We show the time taken to unlearn by each of the methods in Figures 7, 8, 9. *Cognac* achieves speed-ups up to 12.5% in some datasets, and the time taken by is competitive with, or often lesser than most baselines.

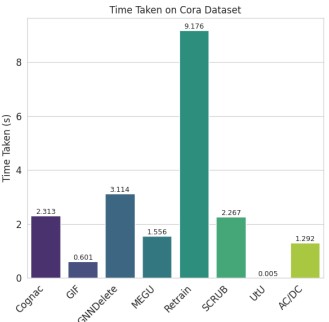

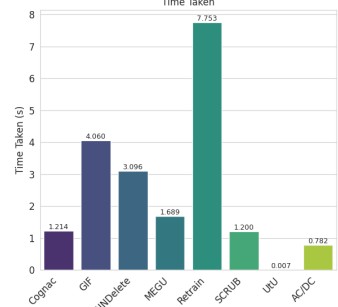

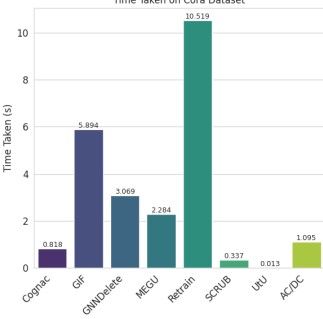

Figure 7: Time taken for unlearning by different methods on the Amazon dataset

Figure 8: Time taken for unlearning by different methods on the CS dataset

Figure 9: Time taken for unlearning by different methods on the Cora dataset

# F PERFORMANCE ON LARGE $S_f$

To stress-test *Cognac*'s performance - as methods could potentially degrade as the size of $S_f$ grows - we conduct an experiment where we choose a significant fraction of the training nodes of the Amazon, DBLP and Physics datasets, to be attacked (by the binary label flip attack) and marked for deletion. The method performs competitively even at this large deletion size. Table 7 demonstrates these results.

Table 7: **Results on Node Unlearning for a large $S_f$ value.** *Cognac* performs well across datasets, even when a significant fraction of the dataset is manipulated and is part of $S_f$. The $S_f$ values listed in parenthesis are relative to the size of the *entire* training set.

| Method | **Amazon** (25%) | | **DBLP** (29.4%) | | **Physics** (14.8%) | |
|---|---|---|---|---|---|---|
| | $Acc_{rem}$ | $Acc_{aff}$ | $Acc_{rem}$ | $Acc_{aff}$ | $Acc_{rem}$ | $Acc_{aff}$ |
| Oracle | 92.9 | 95.6 | 72.9 | 86.0 | 95.2 | 95.1 |
| Original | 92.5 | 49.0 | 74.9 | 57.9 | 58.9 | 95.4 |
| *Cognac* | **92.4** | **83.7** | **82.3** | 81.7 | **90.7** | **95.0** |
| GNNDelete | 27.7 | 49.7 | 45.0 | 49.6 | 1.4 | 37.3 |
| SCRUB | 92.3 | 72.6 | 77.5 | **82.1** | 77.9 | 94.9 |

## G  PERFORMANCE ON ADDITIONAL DATASETS

We have additionally tested *Cognac* and the best-performing baselines on DBLP (Tang et al., 2008), Physics (Shchur et al., 2019), Computers (Shchur et al., 2019), and the OGB-arXiv (Wang et al., 2020) datasets using the binary label flip attack (more details about the datasets in Table 8). Similar to the standard GCN performance for OGB-arXiv reported by Kipf & Welling (2017), the Oracle achieves a micro accuracy of $71.15\%$ on the overall test set. As in the rest of the paper, the reported values in Table 9 are macro-accuracies. As shown in Table 9 and Figure 10, *Cognac* is still consistently the top performer on maintaining $Acc_{aff}$ on all the new datasets and even matches the performance of the Oracle model on DBLP and Computers while maintaining $Acc_{rem}$ across datasets and deletion sizes.

Table 8: Additional Dataset Details

| Dataset | # Classes | # Nodes | # Edges |
|---|---|---|---|
| DBLP | 4 | $17,716$ | $105,734$ |
| Physics | 5 | $34,493$ | $495,924$ |
| Computers | 10 | $13,381$ | $245,778$ |
| OGB-arXiv | 40 | $169,343$ | $1,166,243$ |

Table 9: **Results on the OGB-arXiv dataset for $S_f/S_m = 0.25, 1.00$.** We observe that the Cognac method outperforms all baseline approaches, including Retrain, in both correcting the $Acc_{aff}$ metric and maintaining the $Acc_{rem}$ metric.

| Method | 0.25 | | 1.00 | |
|---|---|---|---|---|
| | $Acc_{aff}$ | $Acc_{rem}$ | $Acc_{aff}$ | $Acc_{rem}$ |
| Oracle | $66.8_{\pm 0.0}$ | $49.2_{\pm 0.0}$ | $66.8_{\pm 0.0}$ | $49.2_{\pm 0.0}$ |
| Original | $40.5_{\pm 0.0}$ | $48.6_{\pm 0.0}$ | $40.5_{\pm 0.0}$ | $48.6_{\pm 0.0}$ |
| Retrain | $54.9_{\pm 5.0}$ | $46.3_{\pm 1.0}$ | $59.1_{\pm 3.9}$ | $44.4_{\pm 2.0}$ |
| *Cognac* | $\mathbf{58.0_{\pm 0.5}}$ | $\mathbf{49.6_{\pm 0.1}}$ | $59.2_{\pm 0.7}$ | $\mathbf{49.7_{\pm 0.1}}$ |
| SCRUB | $54.5_{\pm 1.1}$ | $49.2_{\pm 0.1}$ | $49.2_{\pm 0.2}$ | $48.9_{\pm 0.1}$ |
| UtU | $38.4_{\pm 0.0}$ | $48.7_{\pm 0.0}$ | $30.4_{\pm 0.0}$ | $48.9_{\pm 0.0}$ |
| GNNDelete | $44.3_{\pm 4.8}$ | $6.0_{\pm 4.2}$ | $27.4_{\pm 4.1}$ | $32.0_{\pm 2.6}$ |

## H  CONVERGENCE

We now discuss the convergence properties of *Cognac*. Plots in Figure 11 describe the losses of each of the components of our method (contrastive, ascent, descent) after the last epoch of every *step*, the meaning of which should be clear from Algorithm 1: Line 4 (where it's called 'num_steps'). The loss plots are constructed over the best hyperparameters, and we would likely not see such convergence trends with sub-optimal hyperparameters, which may provide insights to improve performance when it's used in other settings as well.

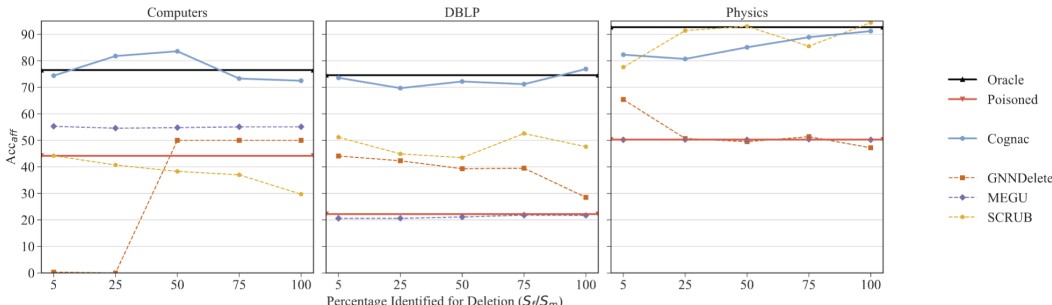

Figure 10: **Evaluation on Node Unlearning for Computers, DBLP, Physics (left to right) datasets**. *Cognac* still outperforms the baselines across the percentages of the deletion set identified and matches Oracle's performance on the new datasets.

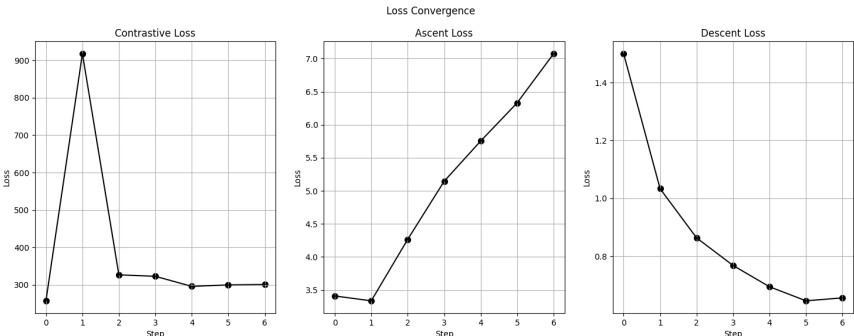

Figure 11: **Convergence of the losses across unlearning steps**. The ascent loss on $S_m$ continually increases as expected, the descent loss on $S \setminus S_m$ converges, and the contrastive loss exhibits a low plateau after an initial overshoot, implying it may have learnt discriminative features.

## I    ANALYSIS OF METHOD USED TO FIND AFFECTED NEIGHBOURS

Our strategy to find affected neighbours is likely not the perfect for finding the most affected nodes and more sophisticated influence functions such as the one presented in Chen et al. (2023) could be used to potentially improve performance. Still, we note that it achieves a $5\%$ higher $\text{Acc}_{\text{aff}}$ than while choosing random $k\%$ nodes in the $n$-hop neighbourhood (where $n$ is the number of layers of message passing) while being cheap to compute: we only require a single forward pass over the model with the inverted features. Interestingly, Figure 12 also shows that even if the GNN is not well-trained, if we choose the top $k\%$ affected nodes, the unlearning performance does not change much, while still being noticeably better than when we use a random $k\%$ of the neighbours.

Figure 13 (left) shows that there are no noticeable changes in taking a smaller or larger $k\%$. However, removing this step entirely ($k = 0\%$) results in worse performance, suggesting performing contrastive unlearning on even a small $k\%$ is significant. Additionally, by keeping this percentage small, we ensure computational efficiency without diminishing performance (Figure 13 (right)), which is essential for unlearning methods.

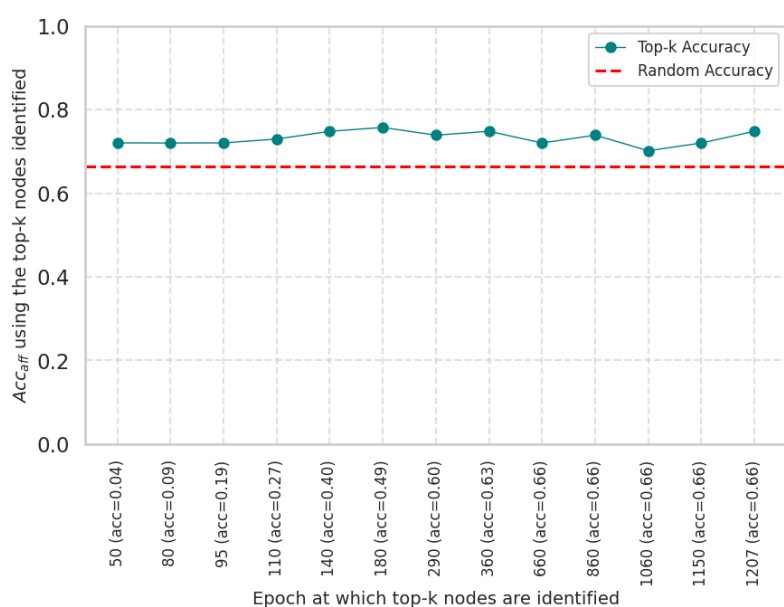

Figure 12: **Effect of how well-trained the GNN is on top-$k\%$ affected neighbour identification.** Here $k = 4$. The x-axis represents the epoch at which we used GNN representations to identify the most affected neighbours for *Cognac*. The y-axis reports the unlearning performance after contrastive unlearning on these identified nodes using the final model. The red line contains performance after picking a random subset from the $n$-hop neighbourhood. Affected neighbourhood identification using top-$k\%$ logit change is more effective even with an extremely undertrained GNN.

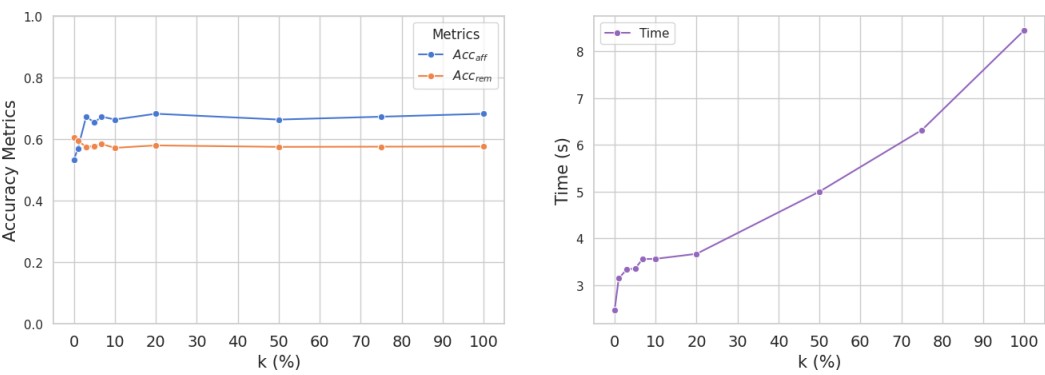

Figure 13: **Effect of $k$ on unlearning ($\mathrm{Acc_{aff}}$), utility ($\mathrm{Acc_{rem}}$) and efficiency when identifying top-$k\%$ affected nodes for contrastive unlearning.** (Left) *Cognac* effectiveness sharply improves beyond $k = 0\%$, suggesting that performing contrastive unlearning on even a small percentage of nodes ($k$) significantly enhances the algorithm's effectiveness. However, using higher values of $k$ yields similar performance with the added downside of increasing computational time (Right).

