# OpenReview forum: "A shot of Cognac to forget bad memories: Corrective Unlearning in GNNs"
_ICLR.cc/2025/Conference — Submitted to ICLR 2025_

### Official Review · Reviewer_bduh · 2024-10-23

**Soundness:** 3
**Presentation:** 2
**Contribution:** 3
**Rating:** 6
**Confidence:** 3

**Summary:**

This paper propose **Cognac** model to tackle corrective unlearning in Graph Neural Networks, efficiently removing the impact of adversarial manipulations, even when only 5% of the manipulation set is identified. It recovers most of the performance of an oracle trained with fully corrected data and is 8x more efficient than retraining from scratch. This makes **Cognac** effective in real-world scenarios with limited knowledge of corrupted data, enhancing GNN robustness post-training.

**Strengths:**

(1) The proposed graph contrastive unlearning is a novel and interesting concept.

(2) The method presented in this paper is simple and effective, without introducing additional discriminative components.

**Weaknesses:**

(1) The experiments in the paper are seriously insufficient. In the graph domain, there are numerous heterophilic and homophilic graphs, such as Wisconsin, Cornell, and Citeseer. The authors only demonstrate results on three graph datasets, which is clearly inadequate.

(2) There are some conceptually similar works [1], and the authors may need to provide a clearer comparison with these related works.

[1] The Heterophilic Snowflake Hypothesis: Training and Empowering GNNs for Heterophilic Graphs

**Questions:**

please refer weakness

---

> ### Author Response · Authors · 2024-11-20
>
> Thank you for the feedback! We are glad that you found the problem novel and interesting, and the proposed method effective. We highlight how we have updated the paper to address the weaknesses **(W)**, and hope this improves the overall rating of our work. We also noticed a lower score in the presentation, please let us know if you have any suggestions for improvement.
>
> ---
>
> **W1:** *The experiments in the paper are seriously insufficient*
>
> We tested our method with 2 adversarial evaluations across 3 datasets, and 5 varying fractions of manipulated data, with several ablations of our method. Overall, we perform **over 3000 runs per unlearning method** to try our different hyperparameters and converge on the best performances for each setting.  As also agreed by reviewers LxzZ and A7wi, we believe this makes our experiments comprehensive for the graph ML domain.
>
> Taking your feedback into account, to increase the breadth of our experiments, we have now added results on the binary label flip attack across three large benchmark datasets used in prior graph unlearning literature (Amazon-Computers [4], Physics [4], DBLP [5]) where we test our method and the best-performing baselines. Details about these datasets are provided in **Table 8**. From **Figure 10**, we can observe that Cognac still outperforms the baselines in most of the settings and matches Oracle’s performance in the new datasets.
>
> ---
>
> **W2:** *Authors may need to provide a clearer comparison with these related works.*
>
> Thank you for the reference, and for allowing us to further clarify the scope of our work. Like most of the previous works in the domain of graph unlearning, such as GNNDelete [1], MEGU [2] and GIF [3], our work is intended for homophilic graphs and consequently caters to architectures like GCN and GAT. These are also the baselines we evaluate against in our work, and thus for a fair comparison, we primarily evaluate on datasets used in these works. We have now acknowledged focusing on homophilic graphs in the **Limitations section L524-525** along with the cited reference, and agree that heterophilic graph unlearning would indeed be an intriguing avenue for follow-up works.
>
> ---
>
> [1] Jiali Cheng, George Dasoulas, Huan He, Chirag Agarwal, & Marinka Zitnik (2023). [GNNDelete: A General Strategy for Unlearning in Graph Neural Networks.](https://openreview.net/forum?id=X9yCkmT5Qrl) In *The Eleventh International Conference on Learning Representations* .
>
> [2] Li, X., Zhao, Y., Wu, Z., Zhang, W., Li, R. H., & Wang, G. (2024, March). [Towards Effective and General Graph Unlearning via Mutual Evolution.](https://ojs.aaai.org/index.php/AAAI/article/view/29273/30403) In *Proceedings of the AAAI Conference on Artificial Intelligence* (Vol. 38, No. 12, pp. 13682-13690)
>
> [3] Wu, J., Yang, Y., Qian, Y., Sui, Y., Wang, X., & He, X. (2023). [GIF: A General Graph Unlearning Strategy via Influence Function.](https://dl.acm.org/doi/pdf/10.1145/3543507.3583521) In *Proceedings of the ACM Web Conference 2023* (pp. 651–661). ACM.
>
> [4] Shchur, O., Mumme, M., Bojchevski, A., & Günnemann, S. (2018). [Pitfalls of graph neural network evaluation](https://arxiv.org/pdf/1811.05868). *arXiv preprint arXiv:1811.05868*
>
> [5] Tang, J., Zhang, J., Yao, L., Li, J., Zhang, L., & Su, Z. (2008). [Arnetminer: extraction and mining of academic social networks](https://dl.acm.org/doi/pdf/10.1145/1401890.1402008). In *Proceedings of the 14th ACM SIGKDD international conference on Knowledge discovery and data mining* (pp. 990–998)

---

> ### Comment · Reviewer_bduh · 2024-11-24
> **Response to authors**
>
> Thank you for your responses, I will raise my score!

---

> > ### Author Response · Authors · 2024-11-24
> >
> > We are glad we could resolve your concerns. Thank you for your feedback in helping us improve the paper!

---

### Official Review · Reviewer_A7wi · 2024-10-29

**Soundness:** 3
**Presentation:** 3
**Contribution:** 3
**Rating:** 5
**Confidence:** 5

**Summary:**

This paper introduces *Cognac*, a corrective unlearning framework designed for Graph Neural Networks (GNNs) to mitigate the adverse effects of manipulated data, even when only a small subset of compromised nodes or edges is identified. Cognac combines two key components: (1) *Contrastive Unlearning on Graph Neighborhoods* (CoGN), which minimizes the influence of manipulated nodes on neighboring nodes via contrastive learning, and (2) *Ascent-Descent Decoupled* (AC DC) gradient optimization, which unlearns harmful labels by alternating between ascent and descent steps. This method effectively reduces adversarial impact while maintaining efficiency and scalability.

**Strengths:**

1. Cognac’s integration of CoGN and AC DC components offers an innovative approach to corrective unlearning in GNNs, utilizing contrastive learning and decoupled optimization to handle the unique challenges posed by graph-structured data.
2. The framework demonstrates strong performance, achieving effective unlearning even when only 5% of manipulated data is identified, thus substantially lowering the typical data requirement for effective unlearning.
3. The experiments are comprehensive, covering various manipulation types, model types, and datasets. The paper’s ablation studies and visualizations provide clear insights into Cognac’s efficacy and the mechanisms behind it.

**Weaknesses:**

1. The evaluation metrics (ACC_aff and ACC_rem) mentioned in Lines 175–184, Page 4 differ from traditional accuracy metrics, which directly assess model performance on the original data. Does this mean that the model’s performance on the original data was not evaluated? Or does it imply that naive unlearning performance metrics were excluded? For clarity and comprehensive understanding, I suggest including results from standard unlearning metrics for comparison, even if they may not be favorable for the proposed method. Additionally, if ACC_aff and ACC_rem are novel metrics introduced by the paper, please clarify their originality, or otherwise include citations clearly defined in Lines 175–184.
2. The paper could benefit from a more detailed discussion on why malicious node additions degrade GNN performance. While related studies are referenced, incorporating recent studies would provide a more robust context [1, 2, 3, 4].
3. Correct the typo “th” to “the” in Line 432, Page 9.

**References for Additional Context**:
- [1] *Graph Robustness Benchmark: Benchmarking the Adversarial Robustness of Graph Machine Learning*, NeurIPS 2021, Datasets and Benchmarks Track.
- [2] *Mitigating Emergent Robustness Degradation on Graphs while Scaling Up*, ICLR 2024.
- [3] *Chasing All-Round Graph Representation Robustness: Model, Training, and Optimization*, ICLR 2023.
- [4] *Characterizing the Influence of Graph Elements*, ICLR 2023.

**Questions:**

See Weaknesses Section.

---

> ### Author Response · Authors · 2024-11-20
>
> Thank you for the constructive feedback! We’re glad you found our experiments comprehensive, and recognized our innovation in uniquely handling challenges of graph unlearning to achieve strong performance. We highlight how we updated the paper to address the weaknesses **(W)**, hoping this improves the contribution and overall rating of our work.
>
> ---
>
> **W1:** *The evaluation metrics (ACC_aff and ACC_rem) mentioned in Lines 175–184, Page 4 differ from traditional accuracy metrics*
>
> The metrics are not novel, and we borrow them directly from the recent work which introduced corrective unlearning [1]. To further clarify, we *do* compute the accuracy over the original test (unseen) datapoints, but we just split it into two parts - one calculated over test data points from the manipulated distribution and one over test data points from the remaining data distribution. For example, in case of the label flip attack, $Acc_{aff}$ measures the accuracy over the two classes that were chosen to have flipped labels and $\mathrm{Acc_{rem}}$ measures the accuracy the remaining $n-2$ classes, where $n$ denotes the number of classes. We have added the following lines to **Section 3 (L180-192)** to clarify the same,
>
> > “The metrics $Acc_{aff}$ and $Acc_{rem}$ were termed ”Corrected Accuracy” $(Acc_{corr})$ and “Retain Accuracy” $({Acc}_{retain})$ respectively by [1]. We chose alternative names to explicitly state which data distribution accuracy is measured on. In **Section 5.1**,  we further specify what the "affected distribution” and “remaining entities” are for the different evaluation types we study.”
> >
>
> [1] Shashwat Goel, Ameya Prabhu, Philip Torr, Ponnurangam Kumaraguru, & Amartya Sanyal (2024). [Corrective Machine Unlearning](https://openreview.net/forum?id=v8enu4jP9B). *Transactions on Machine Learning Research.*
>
> ---
>
> **W2:** *The paper could benefit from a more detailed discussion on why malicious node additions degrade GNN performance.*
>
> We have now added further context on graph robustness to include literature on node injections and robust pretraining as suggested, including citations mentioned in the review in **Section 2 (L110-115)**. Thank you for pointing out these relevant papers!
>
> ---
>
> **W3:** *Correct the typo “th” to “the” in Line 432, Page 9.*
>
> We’ve now fixed this typo, and have gone through the manuscript to check for other typos too. Thank you for pointing this out.

---

> > ### Author Response · Authors · 2024-11-25
> > **We hope we clarified your concerns**
> >
> > As the end of the discussion period is approaching, we wanted to check in to make sure we sufficiently clarified your concerns regarding the evaluation metrics and related works. Please let us know if you have any remaining feedback!

---

### Official Review · Reviewer_LzxZ · 2024-10-30

**Soundness:** 2
**Presentation:** 3
**Contribution:** 2
**Rating:** 5
**Confidence:** 3

**Summary:**

This work study the corrective unlearning problem for GNNs, where model developers try to remove the adverse effects of manipulated training data from a trained GNN, with realistically only a fraction of it identified for deletion. The authors introduce a new graph unlearning method, Cognac, which can unlearn the effect of the manipulation set even when only 5% of it is identified.

**Strengths:**

1. The authors provide a coherent story.
2. Empirical studies on the benchmark dataset are comprehensive.

**Weaknesses:**

1. When identifying the affected nodes, the inverse features need to be computed. What is the inverse features? The authors then choose the top K% nodes with the largest logit change. How will the chosen nodes influence the results? I suggest the authors provide empirical analysis of the K%.
2. When calculating the logits change, should the GNN be trained well? I suggest the authors provide more analysis of how the GNN's performance influence on the logits change.
3. The algorithm has two optimization steps. How about the convergence of the algorithm? How to set the parameters of the optimizer?
4. The proposed method lacks justification. I suggest the authors theoretically analyze why selecting the top K% nodes with largest logit change and using the contrastive unlearning objective work.

**Questions:**

See the weaknesses.

---

> ### Author Response · Authors · 2024-11-20
>
> We are glad the reviewer found our experiments comprehensive and message coherent. We thank them for the questions and ideas for ablating the affected node identification part of our method. We address the weaknesses (W) below, and highlighted how we updated our paper to incorporate your suggestions. We hope this improves the soundness, contribution and overall rating of our work.
>
> ---
>
> **W1:**  *What is the inverse features?*
>
> Thank you for pointing out this omission. We have now made the definition of ‘inverse’ features clearer in the paper **(Section 4.1 L219-222)**. We define ‘inverse’ features as: $\vec{1} - X_v$, where $X_v$ is a one-hot encoding feature vector of all $v \in S_m$.
>
> *How will the chosen nodes influence the results? I suggest the authors provide empirical analysis of the $k$%.*
>
> We investigate the impact of selecting top $k$% nodes in our experiment, as detailed in **Figure 13 (left), Appendix I**. While varying $k$ % did not significantly change results, completely removing the step ($k=0$) degraded the performance. This suggests that performing contrastive unlearning on even a small percentage of nodes has a significant impact. Moreover, maintaining a low $k$% ($k \in [2,30]$) ensures computational efficiency without compromising performance, as shown in **Figure 13 (right)**, which is particularly important for unlearning methods.
>
> ---
>
> **W2:** *When calculating the logits change, should the GNN be trained well? I suggest the authors provide more analysis of how the GNN's performance influence on the logits change.*
>
> Thank you for the suggestion. We have added another experiment to clarify this, with results presented in **Figure 12, Appendix I**. We used the top $k$% (fixed at $4$%) most affected nodes from previous checkpoints as anchors on the fully trained GNN, and observe that the GNN's training stage did not significantly influence the $Acc_{aff}$. However, if we select $4$% nodes randomly, we find a drop in performance. This points to the robustness of using logits change (computed at any model checkpoint) as a measure to identify potentially influenced nodes, as it leads to good downstream unlearning performance.
>
> ---
>
> **W3:**  *How about the convergence of the algorithm? How to set the parameters of the optimizer?*
>
> As per your suggestion, we have added **Figure 11 under Appendix H**, where we plot the loss after the last epoch of each step, where step is as defined in **Algorithm 1** in **Appendix B**. The ascent loss on $S_f$ **(Figure 11 middle)** continually increases which is an expected and intended behaviour with gradient ascent. On the other hand, the descent loss on $S \setminus S_f$ **(Figure 11 right)** and the contrastive loss **(Figure 11 left)** converge.
>
> We supply the Adam optimizer for each unlearning component (contrastive, ascent, descent separately) with the following hyperparameters:
>
> - Learning rate: Varies across settings based on hyperparameter search.
> - Weight decay: Part of the hyperparameter search, but we see the optimal value is always found to be $10^{-6}$,
> - Adam specific parameters: $\beta_1$$\beta_2$, $\epsilon$. Used the default values $0.9$, $0.999$, $1e-08$ respectively.
>
> Apart from this, all tunable hyperparameters for our method are explained in the [README.md](http://readme.md/) in the code (which will be made public upon acceptance) provided in the supplementary, along with the exact hyperparameters for reproducibility.
>
> ---
>
> **W4:**  *I suggest the authors theoretically analyze why selecting the top K% nodes with the largest logit change and using the contrastive unlearning objective work.*
>
> We agree that a lack of theoretical guarantees is a weakness of our work, and have now added a line to mention this in the **Limitations, L534-535**.
>
> We use the largest logit changes to identify affected nodes, though more sophisticated techniques like influence functions **[1]** could improve performance. As a sanity check, we note that it still achieves higher $Acc_{aff}$ than selecting random $k$% nodes in the n-hop neighbourhood **(Figure 12)**.
>
> We incorporated contrastive unlearning for its unsupervised nature, which makes our method usable without labels and counteracts the effect of fine-tuning on incorrect labels when the manipulated data is only partially discovered. Leveraging contrastive learning's well-established theoretical framework **[2]**, we adapt the technique to push manipulated samples away from clean neighbours in the representation space (**Figure 3**).
>
> ---
>
> **[1]** Zizhang Chen, Peizhao Li, Hongfu Liu, & Pengyu Hong (2023). [Characterizing the Influence of Graph Elements](https://openreview.net/forum?id=51GXyzOKOp). In *The Eleventh International Conference on Learning Representations*.
>
> **[2]** Arora, S., Khandeparkar, H., Khodak, M., Plevrakis, O., & Saunshi, N. (2019). A Theoretical Analysis of Contrastive Unsupervised Representation Learning. *CoRR*, *abs/1902.09229*. Retrieved from http://arxiv.org/abs/1902.09229

---

> > ### Author Response · Authors · 2024-11-25
> > **We hope we clarified your concerns**
> >
> > As the end of the discussion period is approaching, we wanted to check in to make sure we sufficiently clarified your concerns regarding the method details with the new experiments we added. Please let us know if you have any remaining feedback!

---

> > > ### Comment · Reviewer_LzxZ · 2024-11-27
> > > **Thanks for rebuttal**
> > >
> > > I thank the authors for the rebuttal. Part of my concerns are resolved.

---

> > > > ### Author Response · Authors · 2024-11-27
> > > >
> > > > Thank you for taking time to read our rebuttal and providing valuable feedback. We would be happy to try and resolve any remaining concerns if you could please specify them, especially as the discussion period has been extended.

---

> > > > > ### Comment · Reviewer_LzxZ · 2024-12-03
> > > > >
> > > > > I feel that this work primarily applies corrective unlearning from existing papers directly to graph setting, lacking deeper insights into the field. I have decided to maintain my score. Thanks for the authors' response.

---

### Official Review · Reviewer_47HJ · 2024-10-31

**Soundness:** 3
**Presentation:** 3
**Contribution:** 2
**Rating:** 5
**Confidence:** 4

**Summary:**

This paper tackles Corrective Unlearning in Graph Neural Networks (GNNs) to counter the spread of manipulated data effects. Existing methods fall short, even with complete knowledge of manipulated data. The proposed method, Cognac, effectively reduces manipulation impacts with only partial information (5% of the manipulated set), restoring most of the ideal model’s performance and surpassing retraining efficiency by 8 times.

**Strengths:**

1. The paper is well-written and easy to follow.
2. This paper studies an interesting problem in the graph domain.
3. The motivation to identify most affected nodes make sense.

**Weaknesses:**

1. The experiments use GCN or GAT as backbones. Suppose we have a two-layer GNN; in that case, a node will only impact its 2-hop neighbors through message passing. This makes it confusing why, in Sec 4.1, the authors propose using output logits change as an indicator to find the most affected node.
2. I am concerned about updating the model with Eq. (2). If we have a large number of nodes to unlearn, I believe gradient ascent on $S_f$ would significantly reduce model utility. Could the authors add an experiment to verify this?
3. The experiments are not comprehensive, as they only focus on small graph datasets.

**Questions:**

please refer to weakness

---

> ### Author Response · Authors · 2024-11-20
>
> We are glad that the reviewer found the paper well-written and the problem interesting. We thank the reviewer for recognizing that existing methods fall short, and that our proposed method is effective. Below we respond to the weaknesses **(W).** We hope the updates we made to our paper improve the contribution and overall rating of our work.
>
> ---
>
> **W1:**  *Suppose we have a two-layer GNN; in that case, a node will only impact its $2$-hop neighbors through message passing.* *This makes it confusing why, in Sec 4.1, the authors propose using output logits change as an indicator to find the most affected node.*
>
> You are correct in mentioning that a node’s impact is limited to its $n$-hop neighbourhood for a n-layer GNN. What we find is that using the most affected subset of the nodes in the n-hop neighbourhood for contrastive unlearning is sufficient, and greatly improves efficiency.  We have clarified that the motivation is efficiency in **L217** of **Section 4**. We have now added analysis on the effect of varying the fraction of the $n$-hop neighbourhood used for contrastive unlearning in **Figure 13, Appendix Section I** shows that using a larger fraction of $n$-hop neighbourhood (apart from the top-$k$% chosen by our method) for contrastive unlearning does not lead to better unlearning performance while adding computational cost.
>
> ---
>
> **W2:**  *I believe gradient ascent on would significantly reduce model utility.*
>
> As suggested, we have added an additional experiment to verify this, with results presented in **Table 7** under **Appendix F**. We set the fraction of nodes to be unlearnt as $25$% (more than $2$x the previous size) of all training nodes on the Amazon dataset and perform unlearning. *Cognac* performs well even at a large deletion size, and is not adversely affected by it. This could be attributed to a tunable learning rate of the gradient ascent component of our algorithm: i.e. if the performance of the method is adversely affected due the destabilisation by the ascent, the learning rate is reduced automatically during the hyperparameter search.
>
> ---
>
> **W3:** *The experiments are not comprehensive, as they only focus on small graph datasets.*
>
> We test our method against $6$ baselines with $2$ adversarial evaluations across $3$ datasets, and $5$ varying fractions of manipulated data, with several ablations of our method. Overall, we perform **over $3000$ runs per unlearning method** to try our different hyperparameters and converge on the best performances for each setting. We have chosen three large, and relatively balanced datasets for our experiments: Cora Full, an extended version of the Cora dataset consisting of $18,800$ nodes and $125,370$ edges (the commonly referenced Cora consists of $2708$ nodes, and $5429$ edges with $7$ classes); the CS dataset with $18,333$ nodes and $163,788$ edges; and Amazon-Photos with $7,487$ nodes and $238,086$ edges. As also mentioned by Reviewer LxzZ and A7wi, we believe this makes our experiments comprehensive for the Graph ML domain.
>
> Nevertheless, we have now added results on the binary label flip attack across three benchmark datasets used in prior graph unlearning literature, (Amazon-Computers [1], Physics [1], DBLP [2]) where we test our method and the best-performing baselines. Note that Physics has almost $500$k edges, $2$x more than our previous biggest dataset. Details about these datasets are provided in **Table 8**. From **Figure 10**, we can observe that *Cognac* still outperforms the baselines in most of the settings and matches the Oracle’s performance in the new datasets.
>
> ---
>
> [1] Shchur, O., Mumme, M., Bojchevski, A., & Günnemann, S. (2018). [Pitfalls of graph neural network evaluation](https://arxiv.org/pdf/1811.05868). *arXiv preprint arXiv:1811.05868*
>
> [2] Tang, J., Zhang, J., Yao, L., Li, J., Zhang, L., & Su, Z. (2008). [Arnetminer: extraction and mining of academic social networks](https://dl.acm.org/doi/pdf/10.1145/1401890.1402008). In *Proceedings of the 14th ACM SIGKDD international conference on Knowledge discovery and data mining* (pp. 990–998)

---

> > ### Comment · Reviewer_47HJ · 2024-11-23
> >
> > Thanks for your reply. I would like to increase my rating to 5. Here are some closing thoughts:
> >
> > W1: I believe that in the original submission, the author didn't mention selecting affected nodes within the n-hop neighbors. This omission represents a fundamental conceptual error.
> >
> > W2: I can understand that in some datasets, it is possible to achieve high classification performance even with a few correctly labeled data samples. However, I believe this idea is difficult to generalize to many other datasets.
> >
> > W3: There are many larger datasets, such as OGB-arXiv.

---

> > > ### Author Response · Authors · 2024-11-28
> > > **Rebuttal response (1/2)**
> > >
> > > Thank you for your response. Below we provide some clarifications and hope they resolve the remaining concerns.
> > >
> > > ---
> > >
> > > **W1:** *the author didn't mention selecting affected nodes within the n-hop neighbors. This omission represents a fundamental conceptual error*
> > >
> > > While both our initial formulation and the codebase explicitly had this condition enforced, we omitted that the affected nodes subset is selected from within the n-hop neighborhood inadvertently. We regret this oversight and appreciate your careful reading that helped us fix it, but we respectfully disagree with its characterization as a fundamental conceptual error. We only somehow forgot to mention the n-hop neighborhood constraint when writing the final paper. Evidence of this is available in the code provided in the supplementary material, **Line (162-164 and 181-186)** in file `trainers/contrascent_no_link.py`, which has not been updated during the rebuttal process.
> > >
> > > ---
> > >
> > > **W2:** *I can understand that in some datasets, it is possible to achieve high classification performance even with a few correctly labelled data samples.*
> > >
> > > Thank you for articulating the concern regarding generalization across datasets for the ascent working on larger deletion sets. We ran new experiments (now added to **Appendix F, Table 7**) on two additional datasets, DBLP and Physics, with large deletion set sizes of 29.4% and 14.8% respectively. We observe that *Cognac* continues to perform consistently well, supporting its generalizability across datasets.
> > >
> > > We are unsure what the reviewer means by *“achieve high classification performance even with a few correctly labeled data samples”* in the context of gradient ascent and large deletion set sizes.
> > >
> > > If the reviewer meant choosing a very large deletion set with only a small number of clean samples remaining in the retain set, we agree that good performance would not be feasible in such a scenario. This would also be an impractical unlearning setting, as the original model's utility would be too low to be useful. Hence, for a reasonable unlearning evaluation, we have:
> > >
> > > 1. Limited our experiments to settings where retraining from scratch ("Retrain") shows reasonable performance (we note that *Cognac* matches or occasionally outperforms it)
> > > 2. Ensured the attack budget is sufficiently large to induce a significant accuracy drop on manipulated classes, necessitating unlearning
> > >
> > > We would also like to clarify that our method does not depend on learning from a few correctly labelled samples. Instead, the expectation is to unlearn well even with only a fraction of the manipulated (incorrect) samples identified. *Cognac* is quite effective at removing the effect of the whole manipulation even when only a fraction of the manipulated (incorrect) samples are provided, and we believe this is quite a significant contribution, as the corrective unlearning paper [1] (published at TMLR very recently, i.e. October 2024) showed this is a hard challenge for existing unlearning methods.
> > >
> > > We hope this clarifies the concern and we are happy to address any follow-up questions if we have misinterpreted the point.
> > >
> > > [1] Shashwat Goel, Ameya Prabhu, Philip Torr, Ponnurangam Kumaraguru, & Amartya Sanyal (2024). [Corrective Machine Unlearning](https://openreview.net/forum?id=v8enu4jP9B). *Transactions on Machine Learning Research.*

---

> ### Author Response · Authors · 2024-11-28
> **Rebuttal response (2/2)**
>
> ---
>
> **W3:** *There are many larger datasets, such as OGB-arXiv.*
>
> We appreciate your suggestion of the OGB-arXiv [1] dataset. We did not have the compute resources to run this earlier, but given the reviewer thinks this point is crucial, procured external compute to expand our experiments to this larger dataset. As shown in **Table 9, Appendix G**, Cognac consistently outperforms all baselines and Retrain from Scratch. For ease, we also include the table below in this response. Moreover, to confirm that our GCN model is well-trained, we checked that the Oracle GCN reaches a micro accuracy of $71.15$% on the OGB-arXiv test set, matching the performance reported by [2]. Note that, due to limitations in time and compute, we were only able to run a subset of the settings on this larger dataset. In the camera ready, we will add results across all settings studied.
>
> We hope this resolves all your concerns and is sufficient to recommend acceptance!
>
> ---
> **Table: Node Unlearning results on the OGB-arXiv dataset for $S_f/S_m \mathbf{= 0.25, 1.00}$.** We observe that the Cognac method outperforms all baseline approaches, including Retrain, in both correcting the $Acc_{aff}$ metric and maintaining the $Acc_{rem}$ metric.
>
>
> | **Method**  | **$0.25 (Acc_{aff}$)**   | **$0.25 (Acc_{rem}$)**   | **$1.00 (Acc_{aff}$)**   | **$1.00 (Acc_{rem}$)** |
> |-------------|---------------------|---------------------|---------------------|---------------------|
> | **Oracle**   | $66.8_{\pm 0.0}$    | $49.2_{\pm 0.0}$    | $66.8_{\pm 0.0}$    | $49.2_{\pm 0.0}$    |
> | **Original** | $40.5_{\pm 0.0}$    | $48.6_{\pm 0.0}$    | $40.5_{\pm 0.0}$    | $48.6_{\pm 0.0}$    |
> | **Retrain**  | $54.9_{\pm 5.0}$    | $46.3_{\pm 1.0}$    | $59.1_{\pm 3.9}$    | $44.4_{\pm 2.0}$    |
> |             |                     |                     |                     |                     | <!-- Empty row -->
> | **Cognac**     | $\mathbf{58.0_{\pm 0.5}}$ | $\mathbf{49.6_{\pm 0.1}}$ | $\mathbf{59.2_{\pm 0.7}}$ | $\mathbf{49.7_{\pm 0.1}}$ |
> | **SCRUB**    | $54.5_{\pm 1.1}$    | $49.2_{\pm 0.1}$    | $49.2_{\pm 0.2}$    | $48.9_{\pm 0.1}$    |
> | **UtU**      | $38.4_{\pm 0.0}$    | $48.7_{\pm 0.0}$    | $30.4_{\pm 0.0}$    | $48.9_{\pm 0.0}$    |
> | **GNNDelete**  | $44.3_{\pm 4.8}$    | $6.0_{\pm 4.2}$     | $27.4_{\pm 4.1}$    | $32.0_{\pm 2.6}$    |
>
> ---
>
> [1] Wang, K., Shen, Z., Huang, C., Wu, C. H., Dong, Y., & Kanakia, A. (2020). [Microsoft academic graph: When experts are not enough.](https://www.semanticscholar.org/paper/Microsoft-Academic-Graph%3A-When-experts-are-not-Wang-Shen/ea9a516d5cb0b298f0df50e82b3e0400b72fcdff) *Quantitative Science Studies*, *1*(1), 396-413
>
> [2] Kipf, T. N., & Welling, M. (2016). [Semi-supervised classification with graph convolutional networks](https://openreview.net/pdf?id=SJU4ayYgl). *arXiv preprint arXiv:1609.02907*

---

> > ### Comment · Reviewer_47HJ · 2024-12-03
> >
> > I appreciate the detailed feedback from the author. However, I agree with reviewer LzxZ, and I believe that the contributions of this paper fall below the acceptance bar.

---

### Author Response · Authors · 2024-11-20
**Summary of Paper Revision**

We thank all reviewers for their constructive feedback.  We have responded to each reviewer individually, and below, we also summarise the updates to our paper including additional results and recommended writing changes.

---

**Section 2 (L110-115, Page 2):** We have added further context on graph robustness to include literature (citing the relevant literature pointed out in a review) on node injections and robust pretraining.

**Section 4 (L219-222, Page 5):**  We have restated the definition of ‘*inverse*’ features for clarity.

**Section 4 (L216-217, Page 5):** We have clarified that the motivation for selecting top $k$% nodes is for efficiency.

**Limitations (L534-535, Page 10):** We acknowledge our work's lack of theoretical guarantees.

**Limitations (L524-525, Page 10):** We have acknowledged that we, like most of the existing graph unlearning literature, focus on homophilic graphs, and have cited the suggested reference for context on heterophilic GNN literature.

**Table 7 (Appendix F, Page 19):** We have added an experiment to show the performance of *Cognac* and the best-performing baselines when $25$% of the training data is manipulated, and observe that *Cognac* still outperforms all baselines.

**Figure 10 (Appendix G, Page 20):** We have added 3 more large datasets, and evaluated *Cognac* and best-performing baselines on the binary label flip attack, and we observe that *Cognac* still outperforms the baselines in most of the settings and matches Oracle’s performance in the new datasets. **Table 8:** provides details about these datasets.

**Figure 11 (Appendix H, Page 20):** We plot and discuss the convergence of the different optimization steps in our algorithm.

**Figure 12 (Appendix I, Page 21):** We observe that the well-trained-ness of the GNN used for getting the top $k$% affected nodes does not influence the $Acc_{aff}$ with any clear trend. As a sanity check, we also show that selecting the top $k$% affected still achieves a higher $Acc_{aff}$ than while choosing random $k$% nodes in the n-hop neighbourhood.

**Figure 13 (Appendix I, Page 21):** We show that using a larger fraction of the $n$-hop neighbourhood (apart from the top $k$% chosen by our method) for contrastive unlearning does not lead to better unlearning performance while adding computational cost.

---

> ### Author Response · Authors · 2024-11-28
> **Summary of Paper Revision (2/2)**
>
> We thank the reviewers for engaging in discussions. Consequently, we have added more details in a revised version, summarised below:
>
> **Table 9 (Appendix G, Page 19):** Added experiment of Node Unlearning on OGB-arXiv dataset as requested. We note that OGB-arXiv is a much larger dataset than benchmarks used in our work earlier, with $169,343$ nodes and $1,166,243$ edges. On this dataset as well,  *Cognac* outperforms all baselines including Retrain from Scratch by a significant margin.
>
> **Section 3 (L180-192):** Clarified that evaluation metrics are borrowed from [1].
>
> **Table 8 (Appendix G, Page 19):** We have added details of additional datasets used in the experiments performed during the rebuttal.
>
> [1] Shashwat Goel, Ameya Prabhu, Philip Torr, Ponnurangam Kumaraguru, & Amartya Sanyal (2024). Corrective Machine Unlearning. Transactions on Machine Learning Research.

---

### Author Response · Authors · 2024-12-02
**Summary of Discussion Phase**

Dear Reviewers and ACs,

Thank you for your thorough reviews. We would be happy to provide any further clarifications. We are delighted to note that reviewers find that:

**Contribution**

- The proposed method, *Cognac*, presents a novel and innovative approach to graph unlearning (Reviewers `bduh`, `A7wi`), which is an interesting, important, and well-motivated problem in the graph domain (Reviewer `47hj`).
- *Cognac* effectively reduces adversarial impact while maintaining efficiency and scalability (Reviewer `A7wi`) while being simple and effective (Reviewer `bduh`).
- *Cognac* restores most of the ideal model’s performance, surpassing retraining efficiency by 8 times (Reviewers `47hj`, `bduh`) even with only 5% identified manipulated data (Reviewers `A7wi`, `bduh`, `47hj`, `LzxZ`) across datasets and manipulation types.

**Soundness**

- The empirical studies are comprehensive, covering various manipulation types, model types, and benchmark datasets (Reviewers `LzxZ`, `A7wi`).
- The ablations and visualisations offers clear insights into *Cognac*'s efficacy and mechanisms (Reviewer `A7wi`).

**Presentation**

- The narrative is coherent (Reviewer `LzxZ`), the paper is well-written and easy to follow (Reviewer `47hj`).

---

The reviewers’ feedback has allowed us to improve our manuscript through added breadth in datasets, ablations, and writing changes. Here is how we resolved their initial concerns:

- **Experiments on 4 new datasets, including large scale** (Reviewers `47hj`, `bduh`), **and** **large deletion set sizes** (Reviewer `47hj`): We ran experiments on Computers, Physics, DBLP, and the much larger OGB-arXiv (1.1M edges). Our results demonstrate that *Cognac*'s advantages hold at scale **(Figure 10, Table 9)**, with robust unlearning even at higher fractions of manipulated samples **(Table 7)**.
- **Affected neighbour selection** (Reviewers `47hj`, `LzxZ`): We added ablations that confirm that our design choices for this, while simple and computationally cheaper, enjoy the same performance as using the entire $n$-hop neighbourhood **(Figure 13)** and robustly work even if the original GNN was under-trained **(Figure 12)**. We also improved the description of neighbourhood selection **(Section 4)**.
- **Positioning with respect to related work** (Reviewers `A7wi`, `bduh`): We clarified that we used established evaluation metrics for corrective unlearning as done in prior work (**Section 3)**. We incorporated studies on graph robustness **(Section 2)**, and acknowledged we focus on empirical performance on homophilic graph datasets **(Section 7)**.
- **Convergence analysis** (Reviewer `LzxZ`): We confirmed different components of our method converge as expected **(Figure 11)**.

---

### Author Response · Authors · 2024-12-04
**Clarifying Contributions**

Dear ACs and Senior ACs,

At the end of the discussion phase, Reviewer `LzXz` wrote that *“this work primarily applies corrective unlearning from existing papers directly to graph setting, lacking deeper insights into the field”.* Before we had a chance to respond, Reviewer `47Hj` mentioned *“I agree with reviewer LzxZ”*, citing this as the reason the paper is *“below the acceptance bar”* of ICLR. We strongly disagree with this characterization due to the following reasons,

1. **There are no existing corrective unlearning methods we could directly apply:** The paper highlighting Corrective Unlearning as a problem was released as a pre-print in February 2024 [[Arxiv]](https://arxiv.org/abs/2402.14015), and was only published in October 2024 [[TMLR]](https://openreview.net/forum?id=v8enu4jP9B). We are unaware of any strong solutions to this problem, and are curious to know which existing papers Reviewer `LzxZ` thinks we directly apply to the graph setting. Our proposed method does not even have a direct analogue for image classification or LLMs where corrective unlearning methods [1, 2] have been proposed before, as it has a graph-specific component (Reviewer `A7wi` characterizes this well). Even for components like Ascent and Descent, which are used in many unlearning methods, we adapt them to our setting by making a modification, which leads to 20% unlearning improvements, as shown in Section 6.3.
2. **To even study corrective unlearning on graphs, we had to make important adaptations:** We could not trivially apply attacks from existing corrective unlearning works on images directly to graph data due to its fundamentally different nature (non-i.i.d structure and different features). So, we formulate two effective attacks tailored for both node and edge unlearning (a concept not applicable to images) inspired by the interclass confusion attack and observe that existing graph unlearning methods perform poorly on unlearning poisons, even when provided with complete knowledge about the manipulated data as shown in Section 6.1.
3. **We achieved unprecedented performance for corrective unlearning (surpassing retraining with 5% manipulated data):** No existing unlearning method (across domains) has been shown to be able to remove interclass confusion style attacks in the corrective setting. Our method, *Cognac*, was not only able to achieve that, but also beat retraining in some cases, which is generally considered a gold standard in unlearning (including graph unlearning) as shown in Section 6.1. *Cognac* is thus not only SOTA by huge margins, but also pushes the frontier of what was thought as possible.
4. **We extensively showed deeper insights for unlearning and dealing with manipulated graph data:** In Sections 6.1 and 6.2, we provide insights beyond establishing *Cognac* as SOTA in graph unlearning. We demonstrate that leveraging negative information about affected nodes can help outperform retraining from scratch, often considered the gold standard. Additionally, we analyze a counterintuitive phenomenon where unlearning performance declines as more manipulated samples are identified, showing in Table 3 that retaining node structure significantly improves unlearning. This highlights a key insight: unlike traditional graph unlearning, removing nodes is not always the optimal approach for addressing manipulations.

We are confident our work is a meaningful contribution to the field. We sincerely hope there is room in ICLR for papers that: a) Are the first to study an important problem of increasing interest (corrective unlearning), in an important domain (graph ML) b) Propose a method that is not only SOTA across datasets and evaluations (See Figures 2, 10, Table 9), but also sometimes beats what is considered the gold standard (retraining) while being 8x more efficient c) Have comprehensive ablations and justification for design choices (as highlighted by Reviewer `A7wi`, and further through incorporating all experiments suggested by Reviewers `LzxZ` and `47Hj`, which only reinforced our final design choices).

---

[1] Stefan Schoepf, Jack Foster, Alexandra Brintrup (2024). [Potion: Towards Poison Unlearning](https://openreview.net/forum?id=4eSiRnWWaF). *Journal of Data-Centric Machine Learning Research.*

[2] Li, N., et al. (2024). [The WMDP Benchmark: Measuring and Reducing Malicious Use with Unlearning](https://proceedings.mlr.press/v235/li24bc.html). *Proceedings of the 41st International Conference on Machine Learning*, in *Proceedings of Machine Learning Research*.

---

### Meta-Review · Area_Chair_3bBv · 2024-12-20

**Metareview:**

**(a) Scientific Claims and Findings:**
The paper introduces Cognac, a novel method designed to mitigate the adverse effects of manipulated or incorrect data in Graph Neural Networks (GNNs). Given that graph data often violates the independently and identically distributed (i.i.d.) assumption, adversarial manipulations can propagate through message passing, degrading model performance. Cognac addresses this challenge by enabling corrective unlearning, effectively removing the influence of manipulated entities from a trained GNN. Notably, Cognac can unlearn the effects even when only 5% of the manipulated set is identified, achieving performance close to an oracle with fully corrected training data. It outperforms retraining from scratch without the deletion set while being eight times more efficient.

**(b) Strengths:**
* Innovative Unlearning Method: Cognac presents a novel approach to corrective unlearning in GNNs, addressing a critical need to mitigate the impact of adversarial manipulations or incorrect data. In comparison to related literature, it has to overcome additional challenges like the fact that samples (nodes) are not iid.
* Efficiency: The method demonstrates efficiency, being eight times faster than retraining from scratch, which can be advantageous in practical large-scale applications.
* Partial Identification Capability: Cognac's ability to effectively unlearn with only 5% of the manipulated set identified showcases its robustness and practicality in real-world scenarios where full identification may not be feasible. In a few cases, Cognac also improves over retraining from scratch.

**(c) Weaknesses:**
* Limited Theoretical Analysis: The paper lacks a comprehensive theoretical foundation explaining why Cognac performs effectively, which could strengthen the understanding and acceptance of the method.
* Clarity on Innovation: What were the specific challenges in transferring unlearning ideas to GNNs? How is the fact that samples are non-iid actually addressed as an algorithmic innovation?
* Motivation: To which degree is computational efficiency a relevant concern for GNNs. For most of the studied datasets, retraining the model from scratch on the cleaned dataset is well feasible. The more interesting question is: How/why could cognac lead to improved generalization performance in comparison? (Yet, this is seldomly the case.)
* Scalability Concerns: While Cognac is more efficient than retraining, the paper does not provide a detailed analysis of its scalability to very large graphs, where this increased computational efficiency would actually be a relevant concern. (Only experiments on ogbn were added during the rebuttal.)
* Comparative Evaluation: The submission would benefit from a more extensive comparison with existing unlearning methods in GNNs to clearly highlight Cognac's advantages and potential limitations on a wider set of datasets. Even though four more datasets (Computers, Physics, DBLP, and the much larger OGB-arXiv) were added during the rebuttal, the experiments fall short in comparison with standard GNN evaluations. A comparison even on a conceptual level could provide an intuition of the main idea why Cognac should succeed/improve over competitors.

**(d) Reasons for Rejection:**
After thorough evaluation, the decision to reject the paper is based on the following considerations:
1. Insufficient Justification: The absence of a detailed theoretical analysis undermines the understanding of Cognac's underlying mechanisms and its effectiveness in corrective unlearning. The algorithmic contribution that overcomes GNN specific challenges should be highlighted more. Furthermore, when does the computational speed-up justify the often inferior performance to retraining from scratch?
2. Scalability Uncertainty: The lack of discussion regarding the method's scalability to large-scale graphs and run-time assessments raises concerns about its applicability in practical, real-world settings.
3. Inadequate Comparative Analysis: The paper does not sufficiently compare Cognac with existing unlearning methods (on a conceptual and empirical level), making it challenging to assess its relative performance and contributions to the field. Experiments for heterophilic cases should be added as well. GATs can also work well in this setting and the authors designed Cognac for this architecture.
Addressing these issues would enhance the paper's contribution and its potential for future acceptance.

**Additional Comments On Reviewer Discussion:**

Not all reviewers engaged in the discussion during the rebuttal. Yet, multiple concerns remained and I assess the summarized weaknesses as sufficient to recommend a rejection at this stage to give the authors the opportunity to further improve their paper.

---

### Decision · Program_Chairs · 2025-01-22

Reject